# Drivers of recent decline in dust activity over East Asia

Chenglai Wu [1], Zhaohui Lin [1], Yaping Shao[2], Xiaohong Liu [3] & Ying Li[4]

It is essential to understand the factors driving the recent decline of dust activity in East Asia for future dust projections. Using a physically-based dust emission model, here we show that the weakening of surface wind and the increasing of vegetation cover and soil moisture have all contributed to the decline in dust activity during 2001 to 2017. The relative contributions of these three factors to the dust emission reduction during 2010–2017 relative to 2001 are 46%, 30%, and 24%, respectively. Much (78%) of the dust emission reduction is from barren lands, and a small fraction (4.6%) of the reduction is attributed to grassland vegetation increase that is partly ascribed to the ecological restoration. This suggests that the ecological restoration plays a minor role in the decline of dust activity. Rather, the decline is mainly driven by climatic factors, with the weakening of surface wind playing the dominant role.

The arid and semi-arid land in East Asia is an important dust source of the Earth[1-4]. Dust activity in this region has decreased greatly since the 1970s[5-8]. The decline has been more remarkable in the last two decades after the dusty period of 2000-2002[9-12]. This decrease in dust activity has been beneficial to Chinese, enabling the People's Republic to focus more on combating the anthropogenic air pollution in the 2010s[13]. However, it is unclear whether the decline will continue, and this has raised the basic question on the cause of the decline. Both climatic factors and human activities may have affected the past dust change, and the confidence in our ability to project future dust change hinges strongly on the reliable attribution of the cause to these possible contributors. For the potential mitigation of dust activity, it is essential to identify the role of mitigation measures that have been taken in this region.

Particularly, several major ecological restoration programs have been implemented in China, including the "Three North Shelterbelt Development Program (TNSDP)" (1978), the "Grain to Green Program (GTGP)" (1999), the "Beijing-Tianjin Sand Source Control Project (TNSDP)" (2001), and the "Returning Grazing Land to Grassland Project (RGLGP)" (2003). These programs have resulted in improved vegetation cover in some dust source regions, as evidenced from satellite observations[14,15]. Despite the synchronous occurrence of reduced dust activity and ecological restoration, there has been a debate on whether the ecological restoration programs have played a critical role in reducing the dust activity[12,16-18]. While it is expected that ecological restoration can reduce the dust activity, to what extent this effect is compared to the broad effect of climate change remains unclear.

For the attribution of the decline in dust activity, a quantitative assessment of the impacts from the various influencing factors is necessary. While several statistical analyses[5-7,11,12] suggest that several factors, including surface wind speed, vegetation cover, and soil wetness, influenced dust variations in the past decades in East Asia, it has been difficult to quantify the contributions of the individual factors as they are intermingled together[18]. For a quantitative assessment, it is vital to rely on numerical models as they represent the non-linear impacts of relevant factors on the movements of dust particles in a physical manner[2-4,8,9]. Forced by meteorology and land surface data, these models can be used to study dust activity over a large region and quantify the contributions from individual factors.

Several studies have used models to understand the long-term variations of East Asian dust[8,19-21], and two of them have further separated the contributions of climatic factors and vegetation cover[19,21]. Mao et al.[19] found that vegetation variations in southeastern Mongolia and central northern China have considerable impacts on East Asian dust activity during springtime of 1982 to 2006. Tai et al.[21] showed that

[1]International Center for Climate and Environment Sciences, Institute of Atmospheric Physics, Chinese Academy of Sciences, Beijing 100029, China. [2]Institute for Geophysics and Meteorology, University of Cologne, Cologne 50969, Germany. [3]Department of Atmospheric Sciences, Texas A&M University, College Station, TX 77845, USA. [4]National Climate Center, China Meteorological Administration, Beijing 100081, China. ✉e-mail: wuchenglai@mail.iap.ac.cn; lzh@mail.iap.ac.cn

the interannual variability and trends of East Asian dust emission are largely shaped by climatic factors, with the vegetation cover playing a secondary but locally important role especially in the semiarid and non-desert regions undergoing rapid land cover change and deserti-fication; Of all the climatic factors, surface wind speed, followed by total precipitation, was the most important meteorological factor controlling dust variability and trends, with the weakening of wind playing the largest role in the overall decline of dust emissions in Taklimakan Deserts and Gobi Deserts during 1982-2010. These two studies focused on the period of 1982 to 2006/2010, but it is still unclear about the roles of climatic factors and vegetation cover in the last two decades when more intensive ecological restorations were implemented[15].

Here we use a physically-based dust emission model (Supple-mentary Fig. 1) forced by the latest meteorology reanalysis and long-term satellite vegetation cover to simulate dust emissions over East Asia during 2001–2017. We focus on dust emission which is the first and foremost component of the dust cycle. In the model, the impacts of surface wind speed, soil moisture, vegetation cover, and snow cover on the dust emission are explicitly represented (see Methods). Other factors (e.g., temperature and precipitation) can indirectly impact the dust emission through affecting these factors. The model is first vali-dated with dust storm records (Supplementary Fig. 2) in terms of spatial and temporal variations. Then through a set of numerical experiments (Table 1), we quantify the contributions of the various factors, including surface wind speed, vegetation cover, and soil moisture, to the strong decline of dust activity in East Asia during 2001 to 2017.

## Results

### Weakening of East Asian dust activity in the last two decades

We used dust storm records collected at the synoptic stations to characterize the dust activity during 2001-2017. Dust storm is defined according to World Meteorological Organization guidelines (Supplementary Table 1), and the number of dust storm days was derived and used to estimate the dust storm frequency (see Meth-ods). Dust storms occur in large regions of northern China (north of 35°N; Fig. 1a), but most frequently in the Tarim Basin and Gobi Deserts with an annual number of dust storm days reaching up to 26 on average over 2001–2017. Dust storms also occur frequently (1–5 dust storm days each year) in the adjacent deserts and sandy lands. There is less than one dust storm day per year on average to the east and the south of the deserts/sandy lands and in southwestern Tibetan Plateau. Dust activity also shows large seasonal variations, with dust storms occurring most frequently in spring and least frequently in late summer and early autumn. This result is generally consistent with previous studies[5,6,11], but our study shows a smaller contrast in dust activity between winter and autumn due to the fact that we focus on more recent decades (Supplementary Fig. 3 and Supplementary Notes).

From 2001 to 2017, dust storm days significantly decreased at most stations (statistically significant at the 0.1 level or $p < 0.1$; Fig. 1c). The decrease happened in most regions of northern China with the largest decreasing rate of 10–16 days per decade. In contrast, an increase in dust storm days occurred over a small portion of the sta-tions ($p < 0.1$) in Northwest China, including the eastern part and northern periphery of the Tarim Basin and central Qaidam Basin. Overall, in the eastern part of the dust source region (Eastern Sources shown in Fig. 1a: 35-49 °N, 94–126.5 °E), averaged annual dust storm days decreased from ~3.5 days in 2001 to 0.2–0.4 days in 2015–2017 (Fig. 1e). Dust emitted from Eastern Sources is mainly transported eastward and southward to the densely populated regions in North, Central and East China[22–24]. The weakening of dust activity led to a substantial decrease in dust pollution in these regions. The decreasing trend of dust activity is also reflected in satellite measurements of dust aerosol optical depth (DAOD) in both the dust source regions and downwind regions[25,26] (also see Supplementary Figs. 4-5).

### Dust emission in East Asia constructed with a physically-based model

Dust emission is an important quantity to characterize dust activity, which can be estimated using numerical models[2,9,19–23]. Here a physically-based dust emission model (DuEM v1; code in Supplemen-tary Software) is used to simulate the dust emission flux for 2001–2017 (see Methods). Our results show that the model reproduces the main dust emission regions, which extend from the Tarim Basin, through the Gobi Deserts, to the sandy lands in Northeast China and North China (Fig. 1b). The model also reproduces the dust emission regions over the southern Tibetan Plateau. The extent of dust emission regions (Fig. 1b) is close to the regions where dust storms are recorded (Fig. 1a). The model simulates the strongest dust emission (>100 g m$^{-2}$ yr$^{-1}$) in the Tarim Basin and western Inner Mongolia, which is consistent with the observed dust storm days. Outside China, the model also simulates strong dust emission over the Gobi Deserts in southern Mongolia, which is consistent with the observations in Mongolia[27]. In addition, the model accurately reproduces the seasonal variations of dust storm days, especially the peak of dust storm days in spring (Supplementary Fig. 6). The spatial and seasonal variations of dust emission are shaped by the climatic factors (surface wind speed, soil moisture, and snow cover) and land surface characteristics (vegetation cover and soil properties), as represented by the dust emission model (details in Supplementary Notes and Supplementary Figs. 7-8). These results indicate that, with the various influencing factors incorporated, the model can reproduce the spatial and seasonal variations of dust emission.

During 2001–2017, the model simulation shows a decreasing trend of dust emission fluxes in most regions of Eastern Sources ($p < 0.1$) (Fig. 1d), which is consistent with the decreasing trend of dust storm days in observations. In the eastern part of the Tarim Basin, the model simulates an increasing trend of dust emission fluxes, but the

## Table 1 | Experiments used in this study

| Exp. name | Simulation period | Surface wind speed | Leaf area index | Soil moisture | Dust emission over Eastern Sources (Tg yr$^{-1}$) | |
|---|---|---|---|---|---|---|
| | | | | | **2001** | **2010–2017** |
| Baseline (All: Wind+LAI + SOILM) | 2001–2017 | Historical | Historical | Historical | 310 | 202 |
| Wind | 2001–2017 | Historical | 2001 | 2001 | 310 | 254 |
| LAI | 2001–2017 | 2001 | Historical | 2001 | 310 | 273 |
| SOILM | 2001–2017 | 2001 | 2001 | Historical | 310 | 281 |

Hourly surface wind speed (in terms of friction velocity) and soil moisture (SOILM) as well as daily leaf area index (LAI) are used. For the experiment with one individual factor set to the values as in 2001, the seasonal variations of this factor are still considered and only its interannual variations are excluded. Also shown in the last two columns are annual dust emission over Eastern Sources for 2001 and 2010–2017, respectively.

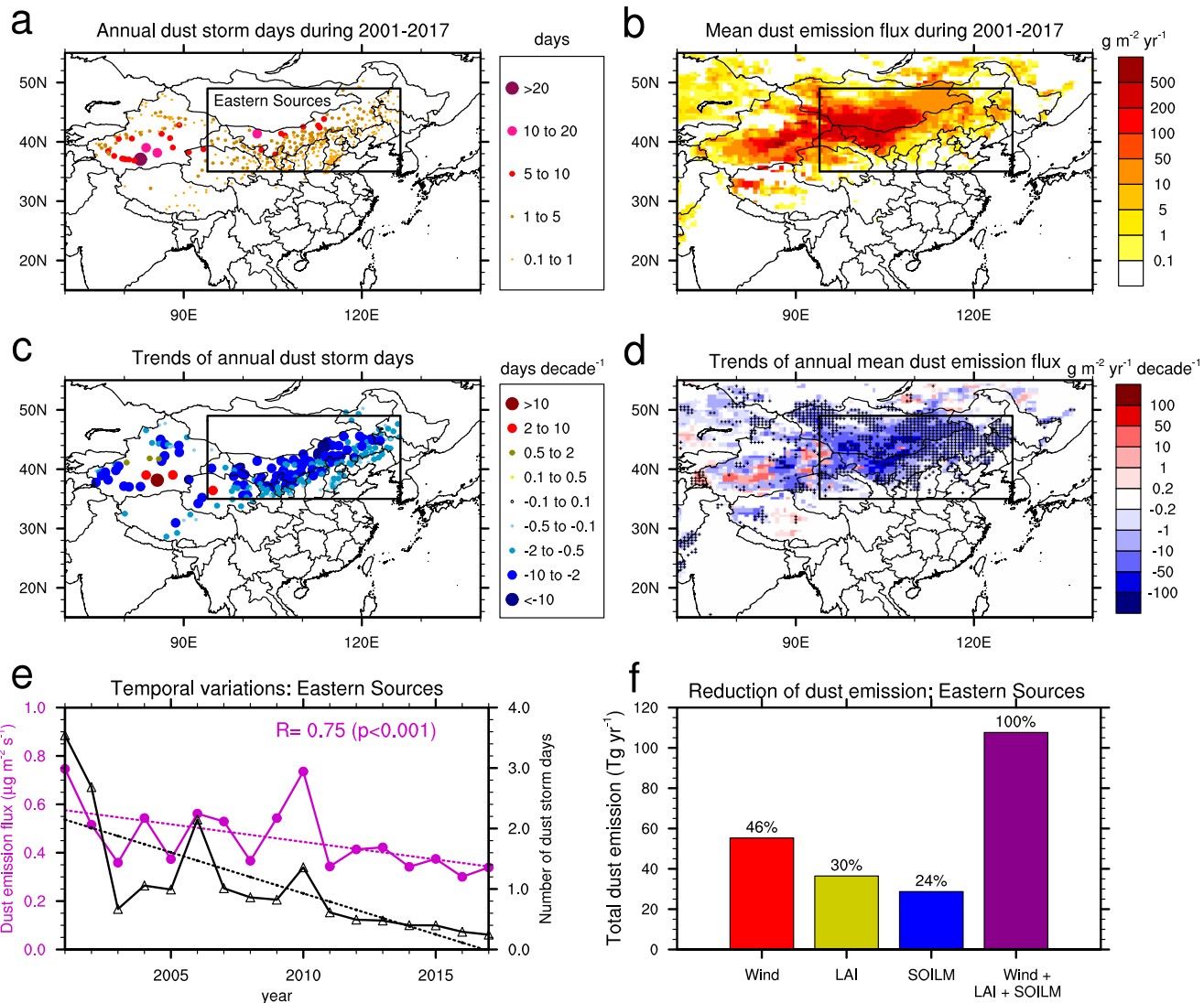

**Fig. 1 | Spatio-temporal variations of observed dust storm days and simulated dust emission flux. a** Annual dust storm days ($N_{DS}$; days) averaged over 2001–2017 and **c** the linear trends of $N_{DS}$ (days decade⁻¹; only stations with $p < 0.1$ are shown). The dust storm days are recorded at 2340 synoptic weather observations. The stations with $N_{DS} < 0.1$ days are not shown in (**a**). **b** Annual mean dust emission flux ($F_{emis}$; g m⁻² yr⁻¹) during 2001–2017 and **d** the linear trends of $F_{emis}$ (g m⁻² yr⁻¹ decade⁻¹; $p < 0.1$ for regions marked with crosses) simulated by the dust emission model (DuEM v1; from the baseline experiment in Table 1) during 2001–2017. The black rectangle denotes the Eastern Sources (35-49 °N, 94-126.5 °E), from which the dust particles are mainly transported eastward and southward to affect the densely populated regions of China. **e** The temporal variations of observed dust storm days (black line marked with triangles) and simulated dust emission flux (magenta line marked with circles; from the baseline experiment) averaged at the synoptic stations over Eastern Sources during 2001–2017. The correlation between two curves is 0.75 ($p < 0.001$). Also shown are the linear trends (dash lines; $p < 0.1$). **f** Reduction of dust emission during 2010–2017 (2010s) relative to 2001, attributable to changes in surface wind ("Wind"), vegetation cover (represented using leaf area index "LAI"), soil moisture ("SOILM") and all these three factors in northern China (from the baseline experiment: "Wind+LAI+SOILM"). These three factors account for 46%, 30%, and 24% of total dust emission reduction. The attribution is derived based on the experiments listed in Table 1.

trend is not as significant as that in the observed dust storm days ($p > 0.1$; Fig. 1d), indicating the air flow complexity related to dust emission in the Tarim Basin may not be well resolved by the MERRA-2 reanalysis due to its relatively coarse resolution[28]. In contrast, in the Eastern Sources, the terrain is smoother and the MERRA-2 meteorology is able to represent the synoptic weather systems, such as cold highs and cyclones, responsible for dust storms.

In the following analysis, we focus on the Eastern Sources. Overall, the decline of dust activity in this area is well reproduced by the model. The simulated mean dust emission fluxes at the synoptic stations decrease significantly ($p < 0.1$) from 0.75 µg m⁻² s⁻¹ in 2001 to 0.34 µg m⁻² s⁻¹ in 2015–2017, and they correlate well with the observed regional mean dust storm days ($R = 0.75$, $p < 0.001$) (Fig. 1e). When using dust emission days (defined as the daily dust emission flux

greater than a critical value such as 1, 10, or 50 µg m⁻² s⁻¹) as an indicator for the comparison with the observations, the correlation between simulations and observations are similar ($R$ ranging from 0.71 to 0.80; Supplementary Fig. 9). In addition, the simulated regionally-accumulated dust emission amount is also well correlated with the regional mean dust storm days ($R = 0.73$, $p < 0.001$) and DAOD ($p < 0.05$), as shown in Supplementary Fig. 10a. These results indicate the dust emission model reproduces well the spatial-temporal evolution of dust emission in Eastern Sources.

**Drivers of the weakening dust activity in last two decades**

The impacts of different factors including surface wind speed, leaf area index (LAI), soil moisture, and snow cover are separately examined. Apart from snow cover, which shows small impacts on the trends of

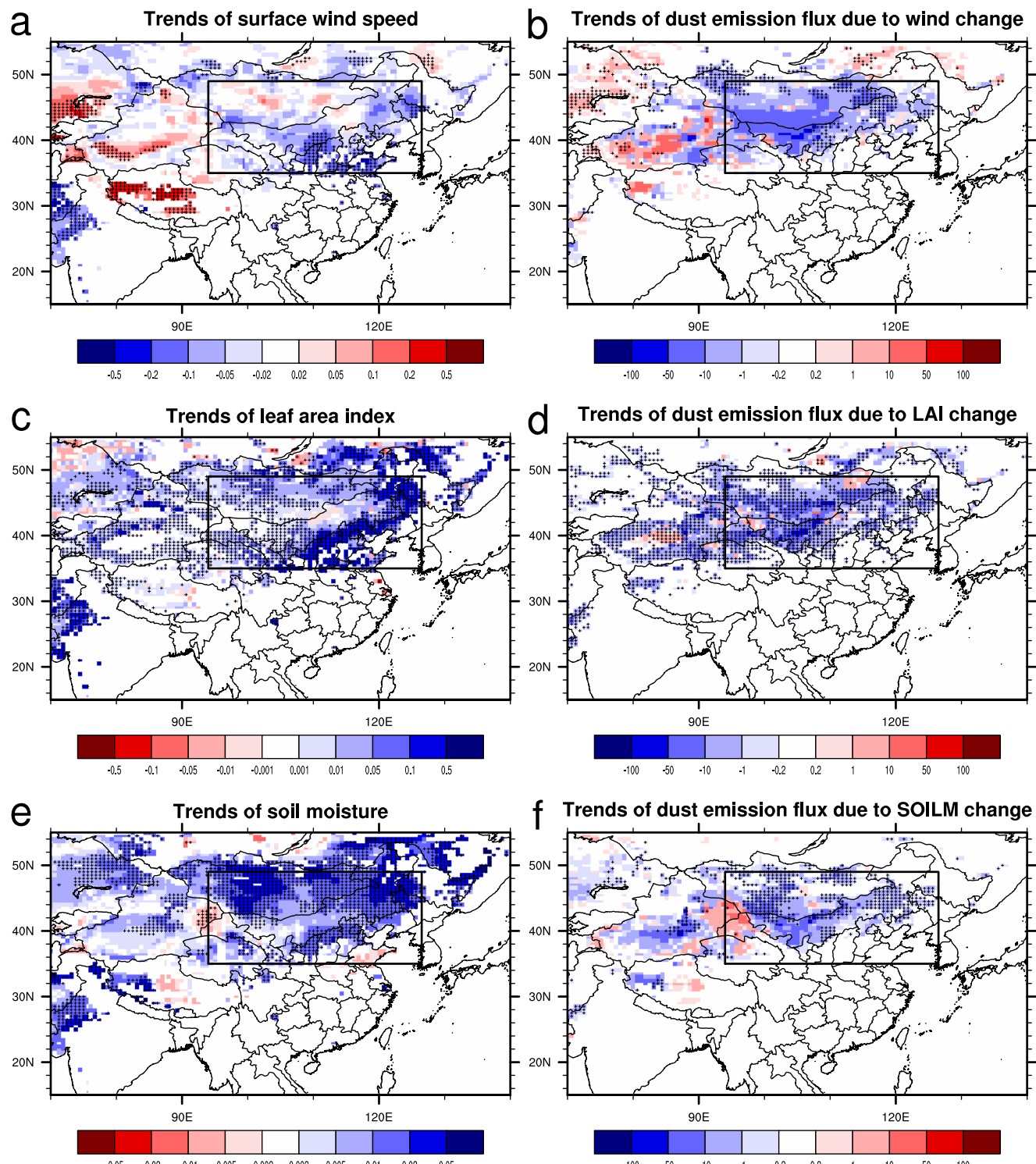

**Fig. 2 | Trends of surface wind speed, leaf area index (LAI), soil moisture (SOILM), and dust emission flux.** Linear trends of (**a**) annual mean surface wind speed (m s⁻¹ decade⁻¹), (**c**) leaf area index (m² m⁻² decade⁻¹), and (**e**) soil moisture (m³ m⁻³ decade⁻¹) during 2001-2017. The trends are shown only in the dust emission regions (mean dust emission flux > 0.001 g m⁻² yr⁻¹). Linear trends of dust emission flux (g m⁻² yr⁻¹ decade⁻¹) during 2001−2017 due to (**b**) changes of surface wind speed (Wind), (**d**) leaf area index (LAI), and (**f**) soil moisture (SOILM), respectively. The crosses denote the regions where the trends are statistically significant (*p* < 0.1). The black rectangle denotes the Eastern Sources (35-49 °N, 94−126.5 °E).

dust emission due to its small change over the main dust source regions during 2001–2017 (Supplementary Fig. 11), other factors contribute significantly to the decline of dust emission in the period from 2001 to 2017 (Fig. 2). Surface winds weakened in most regions of Eastern Sources, leading to the largest reduction in dust emission in this region (Fig. 2a, b). In contrast, surface winds intensified in the

Tarim Basin, causing an increase of dust emission there. Note that due to the large interannual variability of surface winds, the weakening of surface wind was significant (*p* < 0.1) only over part of Eastern Sources–and the same was true for the strong wind event frequency (Supplementary Fig. 12)–but there were larger regions with significant decrease of dust emission (*p* < 0.1). This can be explained by the fact

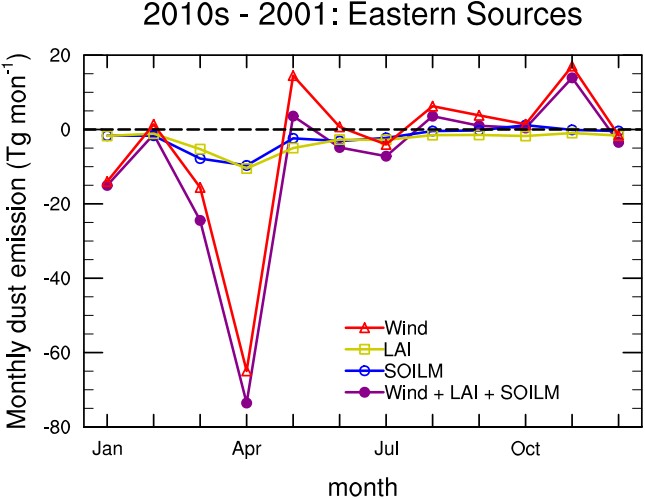

**Fig. 3 | Monthly dust emission change during 2010−2017 (2010s) relative to 2001.** Changes due to individual factors (surface wind–"Wind", vegetation cover–"LAI", soil moisture–"SOILM") and all the three factors together are depicted in red, yellow, blue, and deep magenta, respectively. The monthly changes are derived by calculating the dust emission change for each month from different model experiments (Table 1), as done in Fig. 1f. Note that summing the monthly changes for all the months yields the annual change shown in Fig. 1f.

that dust emission flux is a nonlinear function of wind speed (approximately in cubic function, Supplementary Fig. 1), and a small decrease of strong wind event frequency can result in a substantial reduction in dust emission.

Vegetation cover, represented here using LAI, also caused significant change in the dust emission (Fig. 2c, d). Vegetation greening (a term used to describe the increase in vegetation) reduced the dust emission in most regions of northern China and southern Mongolia. Dust emission was enhanced in a small part of the regions due to vegetation degradation (or browning), such as in the eastern Tarim Basin, the southeastern Mongolia and adjacent China-Mongolia border regions, and a small region of the Qaidam Basin.

Soils became wetter in most regions of northern China, leading to a large reduction of dust emission (Fig. 2e, f). In contrast, soils became dryer in eastern Xinjiang, southeastern Xinjiang, and northwestern Gansu, where dust emission intensified. However, the magnitude of dust emission increase was much smaller than that of dust emission decrease and the net effect was a decrease of dust emission in northern China.

In total, the regional accumulated dust emission decreased by 108 Tg yr$^{-1}$ from 310 Tg yr$^{-1}$ in the first year (2001) to 202 Tg yr$^{-1}$ in the 2010s (2010–2017). Changes in surface wind, vegetation cover, and soil moisture solely accounted for 56 Tg yr$^{-1}$, 37 Tg yr$^{-1}$, and 29 Tg yr$^{-1}$, respectively (Fig. 1f). Note that due to the non-linear effects induced by the combination of different factors, the sum of these individual changes (122 Tg yr$^{-1}$) is larger than that caused by the simultaneous change in the three factors (108 Tg yr$^{-1}$). By dividing the changes caused by each of the factors to the total change (122 Tg yr$^{-1}$), we derived the relative contribution from each factor and found that the changes in surface wind, vegetation cover, and soil moisture accounted for 46%, 30%, and 24%, respectively, of the total change.

The leading role of surface wind speed is also evident in the interannual variations of dust emission, which were mainly explained by surface winds (Supplementary Fig. 10b). In contrast to surface wind speed, vegetation cover and soil moisture caused much weaker interannual variations of dust emission.

Changes of dust emission show large seasonal variations. The decrease in dust emission is most pronounced in spring when dust emission is highest among the four seasons (Fig. 3 and Supplementary Figs. 13-14), particularly in April as dust activity reaches its annual peak (Supplementary Fig. 6). All factors contributed to the decline of dust emission in March and April, but surface wind dominated the change. Surface wind change also led to the dust emission reduction in winter (mainly in January), but it caused considerable increases of dust emission in May, August, and November. The positive change of dust emission canceled out a part of its negative change to yield a net decrease of annual dust emission due to wind speed change. Dust emission reduction due to vegetation cover change was much weaker than that due to surface wind change, but it constantly suppressed the dust emission in most regions for all months, producing a considerable reduction of annual total dust emission (Supplementary Figs. 15-18). While soil drying enhanced dust emission in some parts of Eastern Sources, soil wetting reduced the dust emissions in a larger part of the region in three of four seasons (winter, spring, autumn) as for the annual mean (Fig. 2e, f and Supplementary Figs. 15–18). In summer, soil wetting caused the reduction of dust emission in most of dust source regions. In total, soil moisture change induced a net decrease of annual total dust emission in Eastern Sources with its magnitude comparable to that induced by vegetation cover change.

The relative contributions of each factor to the total changes are summarized in Supplementary Table 2. In spring, the changes in surface wind, vegetation cover, and soil moisture account for 62%, 19%, and 19%, respectively, of the total change. In winter, the relative contributions from these three factors are similar to those in spring. In contrast, the changes in both vegetation cover and soil moisture contribute more to the total changes than surface wind speed in summer, when surface wind speed is lowest during the year.

### Dust emission and changes over different land cover types

Dust source regions in East Asia can be classified into different land types including barren lands, grasslands, and croplands (Supplementary Table 3 and Supplementary Fig. 19). Dust emission in Eastern Sources is mainly from barren lands (Supplementary Fig. 20), which accounts for 84% of the total dust emission (Fig. 4a). Grasslands and croplands account for 15% and 0.7% of the total dust emission in this region, respectively (Fig. 4a). For the dust emission reduction from 2001 to the period 2010–2017 (the emission reduction written as $E_R$ hereafter), barren lands and grasslands accounted for 78% and 20%, respectively (Fig. 4b). Since barren lands are mostly wild, dust emission from there is little affected by direct human activities. Note that 31% of $E_R$ in barren lands is explained by increased vegetation cover (Fig. 4c), suggesting high sensitivity of dust emission to vegetation cover in areas where vegetation is very sparsely distributed. For grasslands, 61% of $E_R$ in these lands was mainly due to surface wind change, and only 23% of $E_R$ in grasslands was a result of vegetation greening (Fig. 4d). As the vegetation greening in grasslands was partly attributed to the ecological restoration programs[15,29], ecological restoration only accounted for a small fraction (less than 4.6%=20%×23%) of $E_R$ in Eastern Sources.

## Discussion

Strong surface winds that generate dust storms are mostly associated with cold air outbreaks and cyclogenesis in the Mongolia Plateau[9,23,30] and thus further influenced by the Westerly Jet, the East Asian Trough, and the Siberian High[7,31]. Vegetation growth depends greatly on climatic factors such as precipitation and temperature[15,29,32]. Soil moisture mainly depends on precipitation and evaporation[33,34] and thus can be affected by temperature and surface wind. Therefore, the impacts of atmospheric circulation, precipitation, and temperature on dust emission are all reflected in the effects of the three factors examined. The weakening of dust activity in the last two decades is mostly attributable to surface wind weakening, vegetation greening, and soil wetting, highlighting the essential role of climatic factors in modulating the dust variability in East Asia.

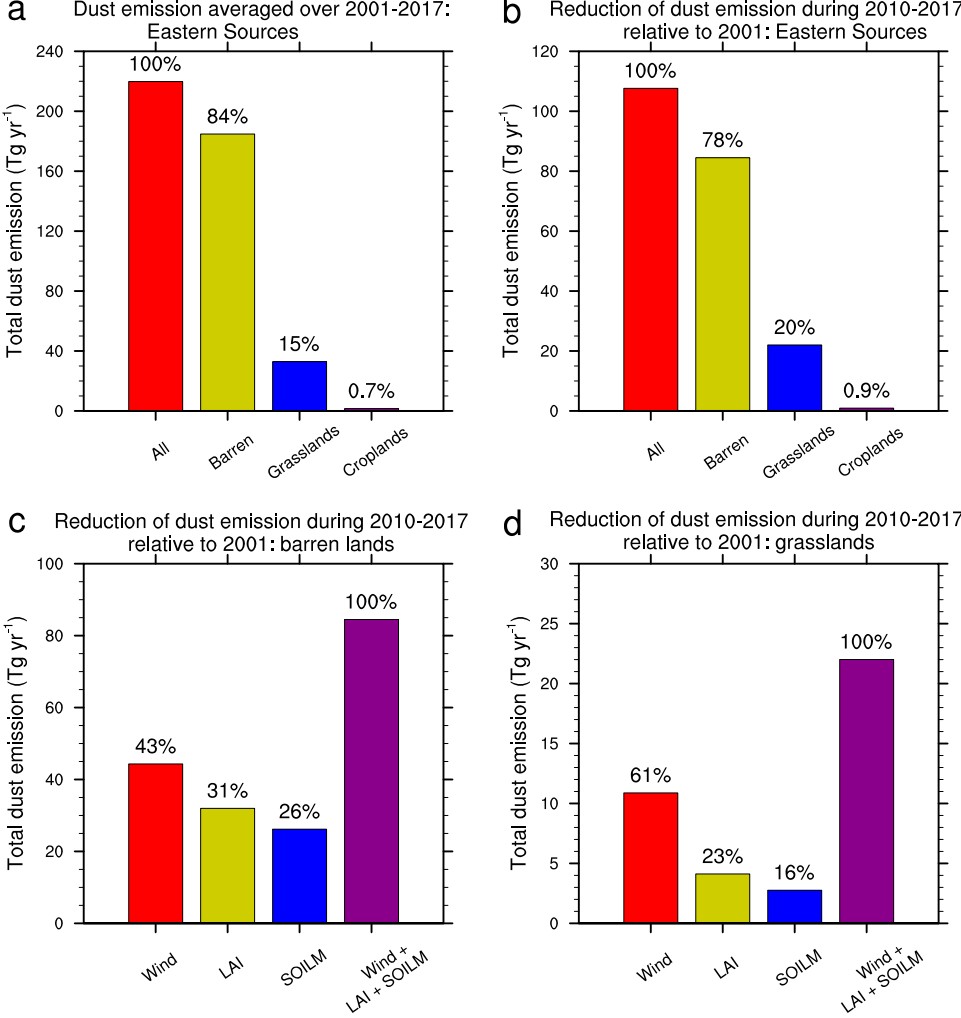

**Fig. 4 | Dust emission and its change over different land cover types. a** Regional accumulated dust emission (Tg yr⁻¹) averaged over 2001–2017 over the whole Eastern Sources and contributions from the barren lands, grasslands, and croplands, respectively. **b** Reduction of dust emission (Tg yr⁻¹) during 2010−2017 relative to 2001 over the whole Eastern Sources and contributions from the barren lands, grasslands, and croplands, respectively. The results shown in (**a**) and (**b**) are from the baseline experiment. Reduction of dust emission (Tg yr⁻¹) during 2010−2017 relative to 2001 over (**c**) barren lands and (**d**) grasslands due to surface wind (Wind), leaf area index (LAI), soil moisture (SOILM), and all these three factors, respectively. For comparison, the relative contributions from different land types (**a**, **b**) or different influencing factors (**c**, **d**) are also provided at the top of each bar.

East Asian dust activity has large seasonal variations and is much stronger in spring than in other seasons. The dust emission reduction in spring accounts for most of the annual total reduction, followed by the reduction in winter (Fig. 3; Supplementary Table 2). Surface winds dominate the dust emission reduction in spring and winter, but changes in surface winds tend to increase the dust emission in autumn and summer. In contrast, changes in vegetation cover and soil moisture lead to the reduction of dust emission in almost all the months (Fig. 3). Although change in either vegetation cover or soil moisture accounts for around 20% of total dust emission reduction by the three factors (surface winds, vegetation over, and soil moisture) in spring and winter, they play a key role in the dust emission reduction in summer (Supplementary Table 2). In autumn, the dust emission reduction induced by vegetation cover greenness cancels out part of the dust emission increase by surface winds. While most of previous studies focused on the dust variations for the spring season[7,19–21,30] or the annual mean[5,6,8,11,18], our study shows large variations in the trends of dust emission and associated surface winds during the four seasons. The mechanisms for these variations deserve further investigation.

Surface winds, vegetation cover, and soil moisture are inter-related to each other. For example, the increase in vegetation cover may decrease the surface winds, which also contributes to the decline of dust activity. Some of this effect may be included in the MERRA-2 data, as the observations are assimilated into the MERRA-2 reanalysis. However, it is difficult to quantify the effect of this interaction on dust emission as the MERRA-2 data are now frozen and no further adjustment is available. Ideally, this limitation can be overcome with a more complex model, such as climate models and Earth System Models (ESMs) which can represent the atmosphere-land-vegetation interactions[8,35]. However, none of the climate models and ESMs to date can reproduce the strong decline of East Asian dust activity during the past decades[8]. Further developments in the forcing data and climate models/ESMs are needed to quantify the impacts of atmosphere-land-vegetation interactions on dust emission.

Note that the previous study of Tai et al.[21] showed that vegetation cover induces much smaller impacts on the long-term variability and trends of dust activity (particularly in Gobi Deserts) compared to climatic factors during 1982-2010. Our study shows larger impacts from vegetation cover change compared to that study, which can partly be ascribed to more intensive change of vegetation cover during 2000s−2010s in our study compared to the earlier decades in that study[15]. Larger impacts in our study may also result from the

differences in the dust emission parameterizations and meteorological forcings (Supplementary Table 4). For example, our model considers the reduction of erodible surface fraction due to increasing LAI, which reduces dust emission, while bare soil fraction is set to be constant in that previous study (Supplementary Table 4).

In this study, we consider the ecological restoration as a major type of human activities that can perturb the land surface and affect the dust emission in large areas. Our model results show only a minor contribution from the ecological restoration to the recent decline of dust activity in East Asia. In addition to the ecological restoration, other human activities may perturb soils directly through cultivation and grazing, which is not considered here. However, the perturbations generally lead to enhanced dust emission by reducing soil cohesion, which may cancel out some of the dust emission reduction due to the ecological restoration.

Although the influence from human activities on dust emission through their perturbations over land surface characteristics is not significant in large areas, other influence of human activities such as anthropogenic greenhouse gas (GHG) emissions and consequent global warming may play a critical role. The weakening of surface wind speed in the mid-latitudes of North Hemisphere can be ascribed to the reduced meridional temperature gradients due to the polar amplification of global warming[36,37] although the magnitude of the effect over the dust source regions is unclear, and a large portion of vegetation greening is also attributable to increased GHG concentrations[15,29,32]. Therefore, anthropogenic influence on the global climate can lead to the decline of dust activity in East Asia indirectly. On the other hand, global warming may induce soil moisture deficit and vegetation loss[38] and thus enhance dust activity, but this effect appears to be weak in the last two decades in Eastern Sources and may be overwhelmed by other factors. In addition to anthropogenic activities, internal variability in the climate system (e.g., the Pacific Decadal Oscillation) can modulate the decadal variations of surface wind speed in East Asia[37,39] and affect dust activity. While global warming is projected to continue in the near future[40], prediction for the phase change in decadal climate variability is difficult and requires further studies.

Despite a small impact on the total dust emission in East Asia, human activity can improve the vegetation cover and reduce local dust emission in some regions[24,41,42]. Not examined here, improved vegetation cover can also regulate local climate and induce negative feedbacks on dust activity[43]. Moreover, the role of human activity may play a more important role if there is a windy year or period in the future owing to the interannual or decadal climate variability. Caution should also be taken when extending our assessment to other dust source regions of the world where the ecological restoration is also of high interest[44], and we suggest that a similar assessment be done as in this study but with the regional characteristics taken into account[45,46]. On the other hand, due to the impacts of dust transport and deposition, the benefits of reduced dust emission in alleviating dust pollution may vary in different regions and should be assessed with the full consideration of dust transport and deposition[20–23]. Overall, this study highlights the combined effects of surface wind speed, soil moisture, and vegetation cover in the decline of East Asian dust activity in recent decades. Therefore, one needs to take into account the evolution of all these three factors to project the future change of dust activity in this region.

## Methods
### Dust emission model
Dust emission strongly depends on near-surface meteorology and land surface conditions (soil texture, soil moisture, roughness, snow cover, etc.)[2]. We use a physically-based dust emission model (DuEM v1; code in Supplementary Software) with these influencing factors incorporated. The model is developed on the basis that the saltation bombardment and aggregate disintegration are the main mechanisms for

vertical dust emission[47,48]. The saltation of soil particles is initiated when the friction velocity, an indicator of wind stress, is greater than the threshold friction velocity which represents the surface resistance for wind erosion[49]. The threshold friction velocity increases due to soil moisture[50] and roughness element[51]. Horizonal saltation flux is proportional to the cube of friction velocity. Based on the soil volume removed by saltating particles, the saltation efficacy (ratio of vertical dust flux and horizontal saltation flux) is calculated by considering the bond strength of soil particles[47,48]. Dust emission only occurred in the exposed bare soil by taking away the fraction of non-erodible surfaces such as snow, vegetation cover, water, etc. The dust emission model performs well when compared with field observations[48,52] and has been widely used in regional[53–56] and global dust modeling[35,57]. The detailed description of dust emission model is provided in Supplementary Methods. Note that vegetation suppresses dust emission in two ways[23]: (1) vegetation cover directly reduces the exposed bare soil; (2) vegetation acts as a major roughness element, consuming part of wind's momentum and increasing the surface resistance (threshold friction velocity). Both effects are explicitly considered in the model. The input variables include friction velocity, air density, soil moisture, snow fraction cover, and vegetation cover, which are prepared from external data; particle size distributions are from field data and assumed invariant. We use this model to calculate the hourly dust emission flux in East Asia during 2001-2017.

### Forcing data
The friction velocity, air density, soil moisture, and snow cover are derived from Modern-Era Retrospective Analysis for Research and Applications, version 2 (MERRA-2), the latest atmospheric reanalysis of the modern satellite era produced by NASA's Global Modeling and Assimilation Office[58]. Instantaneous two-dimensional fields were provided hourly at a resolution of 0.5° × 0.625°. Soil moisture in MERRA-2 is generated with precipitation assimilated and well validated against in-situ measurements[59,60]. Soil moisture is provided at the top layer of 0–5 cm. As the soils subject to dust emission are at the topmost layer of 1-2 cm thickness, a factor of 0.5 is used to scale soil moisture for used in dust emission model[61]. MERRA-2 is provided from 1980 onwards and the data during 2001–2017 are used here. We select the period of 2001–2017 because the Moderate Resolution Imaging Spectroradiometer (MODIS) observation of leaf area index (LAI) is available from July 2000 and well-compiled synoptic records of dust storm are available before 2017, as will be described below. In addition, there are also advantages in focusing on the period of 2000s–2010s when much more observations are assimilated in MERRA-2 compared to the period of 1980s–1990s[58].

LAI during 2001-2017 is derived from Global LAnd Surface Satellite (GLASS) dataset, generated from MODIS reflectance based on general regression neural networks (GRNNs)[62]. MODIS reflectance is first preprocessed to remove the cloud contamination for its input into GLASS algorithm[63]. Then GLASS uses pre-processed reflectance data from an entire year to train the GRNNs and to estimate the one-year LAI profile for each pixel. This method has advantages compared to other existing neural network methods that use only satellite data acquired at a specific time to retrieve the LAI. GLASS LAI shows much improved accuracy compared to original MODIS retrieval[62]. GLASS LAI is originally at a resolution of 0.05° and aggregated into the resolution 0.5° × 0.625° as MERRA-2 using National Center for Atmospheric Research (NCAR) Common Language program (http://www.ncl.ucar.edu/Document/Functions/Built-in/area_hi2lores.shtml). GLASS LAI is provided at 8-day interval and linearly interpolated to generate daily fields.

### Model experiments
To isolate the impacts of different factors, we conducted four model experiments (Table 1). The baseline ("All") experiment uses all the

historical fields of surface wind speed, vegetation cover, and soil moisture. Other three experiments only consider the interannual variations of one factor (such as wind, LAI, and soil moisture) but with other factors fixed as in 2001. Note that for the experiments with the factors fixed as in 2001, the seasonal variations of these factors are still considered and set to the values as in 2001, but the interannual variations of these factors are excluded. By comparing these experiments, we identified the individual contributions of surface wind, vegetation cover, and soil moisture to the interannual variations of dust activity in East Asia.

A recent study has adopted a similar approach to isolate the impacts of climatic factors and vegetation cover[21]. In that study, the impacts of soil moisture and surface wind were considered together. Our study makes a step forward by separating the impacts of these two factors.

Note that although MERRA-2 official dataset also provides the hourly dust emission flux along with meteorological fields, dust emission flux generated by our model agrees better with the observations than that from MERRA-2 official dataset (Fig. 1 and Supplementary Figs. 6 and 21-22). MERRA-2 official dataset uses the scheme of Ginoux et al.[64] which only calculates the dust emission over deserts. Therefore, the official output of MERRA-2 fails to capture the large dust emission regions in central-eastern part of Inner Mongolia, western part of Northeast China, and northern part of North China (Supplementary Fig. 21). It also simulates a weaker seasonal variation of dust storm frequency in Eastern Sources (Supplementary Fig. 22). Our model not only simulates dust emission over deserts, but also over other sparsely-vegetated lands with bare soils such as grasslands and croplands (Supplementary Figs. 19-20) where there are significant seasonal variations in vegetation cover (Supplementary Figs. 7c and 23). Therefore, our model reproduces better the spatial distributions and seasonal variations of dust emission in East Asia than MERRA-2 official dataset. With all influencing factors incorporated in our model, the impacts of these factors including vegetation are isolated through the multiple experiments, which cannot be achieved with MERRA-2 as MERRA-2 data are frozen and no sensitivity experiments (as in our study) can be done.

To separate the dust emission over different land cover types, we also used Collection 6 MODIS Land cover Product (MCD12C1; https://lpdaac.usgs.gov/products/mcd12c1v006/). The land cover fraction based on International Geosphere-Biosphere Programme (IGBP) classification scheme is produced at annual time steps and is used here (Supplementary Table 3). MCD12C1 data is provided at a resolution of 0.05° and aggregated into the resolution of 0.5° × 0.625° as MERRA-2 using NCAR Common Language program (http://www.ncl.ucar.edu/Document/Functions/Built-in/area_hi2lores.shtml).

**Dust storm records at synoptic stations**
We used dust storm (DS) records at the 2340 synoptic stations to characterize the temporal-spatial distribution of dust activity in East Asia. At every three hours, weather phenomenon is recorded, and the weather codes of 09 and 30–35 denote the occurrence of dust storm (Supplementary Table 1). In addition to dust storms, two other kinds of dust events such as dust-in-suspension (DIS) and blowing dust (BD) are also recorded and denoted by the weather codes of 06 and 07, respectively. The definition is made by the World Meteorological Organization (WMO) and commonly used across the world[2,65,66]. According to their definition, DIS is associated with dust transport from other regions or dust suspension after local emission, while BD and DS indicate the occurrences of dust emission. Compared to BD associated with weak and localized dust emission, DS indicates the stronger dust emission that is of regional significance. If there is at least one record of dust storms for a certain day, the day is counted as a dust storm day and a monthly number is derived by summing the dust storm days in the month. Dust emission flux is not directly recorded in

a large region, and dust storm occurrence is used as a proxy for dust emission intensity to validate model simulation here.

According to their geographical locations, the dust sources are divided into two regions (Supplementary Fig. 2): Western Sources, consisting mostly of Tarim Basin (35–44°N, 74–94°E), and Eastern Sources (35–49°N, 94–126.5°E). Eastern Sources include the Gobi Deserts and other deserts and sandy lands in northern China (Supplementary Fig. 2). Dust particles in Tarim Basin are mainly transported westward due to the blocking of surrounding mountains[28,31], while dust particles in Eastern Sources are mostly transported eastward and southward[22–24]. In total, there are 72 and 792 stations in Western Sources and Eastern Sources, respectively. Of these stations, 60 (Western Sources) and 339 (Eastern Sources) stations having more than one dust storm day during 2001–2017 were used here. Note that the synoptic observations cover the period of 1950–2017, and only the observations for 2001–2017 that is overlapped with the period for MODIS LAI data and our model simulations are used here. Regional mean dust storm days were calculated in these regions and compared with the simulations (Fig. 1e and Supplementary Figs. 6, 9, 10a, and 22).

## Data availability
MERRA-2 data is available at https://disc.sci.gsfc.nasa.gov/datasets/. GLASS LAI is available at http://www.glass.umd.edu/Download.html. MCD12C1 data is available at https://lpdaac.usgs.gov/products/mcd12c1v006/. Dust storm records at the synoptic stations and simulated dust emission fluxes are available at https://doi.org/10.57760/sciencedb.03301.

## Code availability
Code for dust emission model is provided in Supplementary Software.

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

## Acknowledgements

This work was supported by the National Key Research and Development Program of China (grant 2020YFA0607801) and the National Natural Science Foundation of China (grant 42075166, 41975119, and 41830966). C. Wu was supported by the Chinese Academy of Sciences (CAS) Pioneer Hundred Talents Program for Promising Youth (Class C). The numerical experiments were conducted at "Earth System Science Numerical Simulator Facility" (EarthLab) which was supported by the National Key Scientific and Technological Infrastructure project.
We thank Dr. Hunter Brown for editing on the early version of the manuscript.

## Author contributions

C.W. and Z.L. designed the study. C.W. did the data analyses. C.W., Z.L., Y.S., X.L., and Y.L. performed the data interpretation and wrote the manuscript. All authors participated in the revision of the manuscript.

## Competing interests

The authors declare no competing interests.
