## [Peer Review File · Nature Communications]

REVIEWER COMMENTS

Reviewer #2 (Remarks to the Author):

General Comments

This paper examines the decreasing trends in dust activity over northern China during the last two decades and quantifies the impact of natural and anthropogenic induced reasons in this declining trend. More specifically, this work quantifies (for the first time) the contribution of wind speed, vegetation cover and soil moisture and their impacts on trends in dust activity over north China from 2001 to 2017. Overall, the decreasing trend in dust activity over the region is known from several previous works, as well as the declining trend in wind speed, but I have not seen any study that quantified the impacts from different parameters. So, from this point of view, the current research has special importance.

In general, it's a well written and organized manuscript, the Supplementary Material is especially rich in describing the model that authors developed for the quantification of the dust fluxes and the impact of each separate parameter, the figures and discussions are of good quality and straightforward and, in general, the manuscript flows well and is easy for the reader to follow it.

I would expect that apart from the trends in dusty days at the meteorological stations, authors to present trends in dust AODs derived either from MERRA-2 or MODIS in order to justify the significant decrease over north China. On the other hand, the literature may be enriched with studies dealing with trends in wind speed over the region, meteorological dynamics associated with wind regimes in north China, studies for environmental restoration projects in arid/desert terrains and so on.

Specific Comments

Line 29: The anthropogenic global warming leads to desertification and decline in moisture and precipitation, which are positive feedbacks for an increase in dust activity. Here, the decline in dust emissions is more related to significant declining trend of wind. So, internal climate variability and mentioning the role of wind is important. I think that the anthropogenic-induced global warming confuses the reader regarding the main finding of this research.

Line 55: "Activity"

Line 72: Add "of dust activity" after "decline"

Lines 210-211: About the wind regime, you may also see and discuss the westerlies over central Asia and the Tarim Basin, which highly modulate dust activity and loess formation over central Asia and NW China. In the respect, you may see a recent work by Li et al. (2021, Geoscience Frontiers) and many references therein. Changes in climatic factors i.e. pressure gradients, the Siberian High and Caspian-Sea High, may highly modulate the atmospheric circulation patterns, wind field and dust activity over the region.

Lines 222-228: However, note that global warming lead to desertification, soil dryness and loss of vegetation which are positive feedbacks for increase in dust activity. This should not be ignored in the discussions here.

In the methods section, refer the period of the MERRA-2 retrievals.

Lines 417-418. The weather codes 06 and 07 also denote dust in suspension and blowing dust (see Hamzeh et al., 2021).

You should modify the colored scale in Fig. 2c, from green to blue colors.

Some recommended literature

- A. Emamian, A. Rashki, D.G. Kaskaoutis, A. Gholami, Ch. Opp, N. Middleton, 2021. Assessing vegetation restoration potential under different land uses and climatic classes in northeast Iran. *Ecological Indicators*, 122, 107325, <https://doi.org/10.1016/j.ecolind.2020.107325>
- H. Gholami, A. Mohammadifar, H. Malakooti, Y. Esmaeilpour, S. Golzari, F. Mohammadi, Y. Li, Y. Song, D.G. Kaskaoutis, K.E. Fitzsimmons, A.L. Collins, 2021. Integrated modelling for mapping spatial sources of dust in central Asia - An important dust source in the global atmospheric system. *Atmospheric Pollution Research* 12, 101173, <https://doi.org/10.1016/j.apr.2021.101173>
- Guan, Q., Yang, J., Zhao, S., (...), Zhang, D., Wu, T. 2015. Climatological analysis of dust storms in the area surrounding the Tengger Desert during 1960–2007. *Clim. Dyn.*
- M., Hamidianpour, S.M.A., Jahanshahi, D.G., Kaskaoutis, A., Rashki, P.G., Nastos, 2021. Climatology of the Sistan Levant wind: Atmospheric dynamics driving its onset, duration and withdrawal. *Atmos. Res.* 260,105711, <https://doi.org/10.1016/j.atmosres.2021.105711>
- N.H. Hamzeh, D.G. Kaskaoutis, A. Rashki, K. Mohammadpour, 2021. Long-term variability of dust events in southwestern Iran and its relationship with the drought. *Atmosphere*, 12, 1350. <https://doi.org/10.3390/atmos12101350>.
- S.K. Kharol, D.G. Kaskaoutis, K.V.S. Badarinath, A.R. Sharma, R.P. Singh, 2013. Influence of land use/land cover (LULC) changes on atmospheric dynamics over the arid region of Rajasthan state, India. *Journal Arid Environments*, 88, 90-101.
- Y., Li, Y., Song, D. G. Kaskaoutis, J., Zan, R., Orozbaev, L., Tan, X., Chen, 2021. Aeolian dust dynamics in the Fergana Valley, Central Asia, since ~30 ka inferred from loess deposits. *Geoscience Frontiers* 101180, <https://doi.org/10.1016/j.gsf.2021.101180>.
- Q., Luo, L., Zhen, Y., Xiao, H., Wang, 2020. The effects of different types of vegetation restoration on wind erosion prevention: a case study in Yanchi. *Environ. Res. Lett.* <https://doi.org/10.1088/1748-9326/abbaff>.
- M., Tian, J., Gao, L., Zhang, H., Zhang, C., Feng, X., Jia, 2021. Effects of dust emissions from wind erosion of soil on ambient air quality. *Atmos. Poll. Res.* [doi:10.1016/j.apr.2021.101108](https://doi.org/10.1016/j.apr.2021.101108).
- J., Xu, Y., Xiao, G., Xie, L., Zhen, Y., Wang, Y., Jiang, 2018. The Spatio-Temporal Disparities of Areas Benefitting from the Wind Erosion Prevention Service. *Intern. J. Environ. Res. Public Health* 15(7):1510, [doi: 10.3390/ijerph15071510](https://doi.org/10.3390/ijerph15071510).

Reviewer #3 (Remarks to the Author):

This study examined the trend in dust storm activity in China over 2001-2017, and used a physical model of dust emission to attribute such trend to different factors, including climatic and vegetation changes. The methods are sound and the results are scientifically valid,

interesting and significant. There are remaining issues that should be addressed before this paper can be accepted for publication, as detailed below.

1. The overall scientific objective and methodology of this study are very similar to the one that was just recently published by Tai et al. (2021) (<https://doi.org/10.1016/j.atmosenv.2021.118348>). In some way this study is an improvement over that by Tai et al., which examined East Asian dust trend over 1980-2010, over which only about 10 years of actual observations are available, and most of their conclusions were based on modeled results. This study has examined nearly 20 years of more recent observations, and thus some of their conclusions were more solidly based on observations. Both studies have found climatic changes and variability to be the major factor shaping East Asian dust emission, while vegetation changes have played some minor roles. There are also some major differences, however. Tai et al.'s study has seen a recent reversal of wind trend and a corresponding increase in dust emission at least over 2000-2010 (which were also documented in the several studies cited by them). They also found that vegetation changes played a much smaller role in controlling dust emission overall than what this study has found. A more comprehensive discussion about the similarities and differences among the two studies is warranted, and a more thorough justification of the novelty of this work should be given.

2. One important difference of this study relative to Tai et al. (2021) is that the model this study used only simulated dust emission, but not the subsequent transport, deposition and thus concentration as did Tai et al. (2021). This makes comparison with observations harder because most of the long-term observations are concentrations or aerosol optical depths, which a dust emission model alone cannot give. This study did compare their simulated dust emission with that provided by MERRA-2; yet, the dust emission product of MERRA-2 is not actual observations, but another model output from their assimilation that also made use of model schemes that were similar to what this study used. Therefore, the model-observation comparison of this study appears to be not entirely based on independent observations; these need to be discussed and addressed in the paper.

3. The authors linked the weakening wind trend to anthropogenic activities that contribute to global warming, and thus related the dust trend to anthropogenic activities in the broader sense. While the general changes in midlatitude wind over large areas can be attributable to global warming, as many studies have found, specifically attributing the past wind changes over the desert and semiarid regions of concern in this study to anthropogenic activities at large requires a lot more climate model experiments in the form of attribution studies than this study has afforded to do. Therefore, the linkage of their changes found, which are solid, to anthropogenic activities as they claimed, is tenuous at best. Therefore, either the authors should tone down this claim, or discuss the connection with much greater details and nuances.

4. The article is at times too colloquial and informal for a scientific paper, e.g., the use of

apostrophes and short forms. These should be avoided, and the article should be rewritten in a more formal tone.

REVIEWER COMMENTS

Reviewer #2 (Remarks to the Author):

General Comments

This paper examines the decreasing trends in dust activity over northern China during the last two decades and quantifies the impact of natural and anthropogenic induced reasons in this declining trend. More specifically, this work quantifies (for the first time) the contribution of wind speed, vegetation cover and soil moisture and their impacts on trends in dust activity over north China from 2001 to 2017. Overall, the decreasing trend in dust activity over the region is known from several previous works, as well as the declining trend in wind speed, but I have not seen any study that quantified the impacts from different parameters. So, from this point of view, the current research has special importance.

In general, it's a well written and organized manuscript, the Supplementary Material is especially rich in describing the model that authors developed for the quantification of the dust fluxes and the impact of each separate parameter, the figures and discussions are of good quality and straightforward and, in general, the manuscript flows well and is easy for the reader to follow it.

I would expect that apart from the trends in dusty days at the meteorological stations, authors to present trends in dust AODs derived either from MERRA-2 or MODIS in order to justify the significant decrease over north China. On the other hand, the literature may be enriched with studies dealing with trends in wind speed over the region, meteorological dynamics associated with wind regimes in north China, studies for environmental restoration projects in arid/desert terrains and so on.

Reply: We thank the reviewer for a detailed review and encouraging comments. The text, figures, and tables are revised as the reviewer suggested.

In particular, we have included the trends of MODIS/Aqua and CALIOP dust aerosol optical depth (DAOD) to justify the significant decrease of dust activity in northern China (Figures S4 and S5). In Figure S4, we can see that DAODs decrease in most regions over Eastern Sources and downwind regions with significant decreasing trends in a large part of the area. In Figure S5, regional mean DAOD has high correlation ($p < 0.005$) with regional mean dust storm days in Eastern Sources, and it also shows a decreasing trend in Eastern Sources ($p < 0.01$ for CALIOP and $p = 0.127$ for MODIS/Aqua). MODIS/Aqua DAOD shows a less significant trend compared to CALIOP DAOD as MODIS/Aqua DAOD began at the year 2003 when lowest dust activity during 2000s was recorded as denoted by dust storm days. Note we do not use DAOD from MERRA-2, as only total AOD is assimilated into MERRA-2 and the ratio of DAOD to total AOD is not constrained in MERRA-2 (Randles et al., 2017).

In the revised manuscript, we have mentioned the decreasing trends of DAOD based on the satellite measurements: **“The decreasing trend of dust activity is also reflected in**

satellite measurements of dust aerosol optical depth (DAOD) in both the dust source regions and downwind regions²³⁻²⁴ (also see Figures S4 and S5).” (Lines 107-110)

In addition, we also added more discussions on the previous studies dealing with the trends in wind speed over the region, meteorological dynamics associated with wind regimes in northern China, environmental restoration in arid/desert regions and other related aspects such as benefits of wind erosion prevent service. Please see the Lines 231-232 (“and thus further influenced by the Westerly Jet, the East Asian Trough, and the Siberian High^{7,29}”), Lines 256-258 (“Not examined here, improved vegetation cover can also regulate local climate and induce negative feedbacks on dust activity⁴⁰”), and Lines 260-266 (“Caution should also be made when extending our assessment to other dust source regions of the world where the ecological restoration is also of high interest⁴¹, and we suggest that a similar assessment be done as in this study but with the regional characteristics taken into account⁴²⁻⁴³. On the other hand, due to the impacts of dust transport and deposition, the benefits of reduced dust emission in alleviating dust pollution may vary in different regions and should be assessed with full consideration of dust transport¹⁹⁻²¹”).

Specific Comments

Line 29: The anthropogenic global warming leads to desertification and decline in moisture and precipitation, which are positive feedbacks for an increase in dust activity. Here, the decline in dust emissions is more related to significant declining trend of wind. So, internal climate variability and mentioning the role of wind is important. I think that the anthropogenic-induced global warming confuses the reader regarding the main finding of this research.

Reply: We thank the reviewer for the comment. We agree with the reviewer that anthropogenic global warming may induce the decline in soil moisture due to either the decrease of precipitation or increase of evaporation. However, based on our study, the decline in soil moisture does not happen in the study region (southern Mongolia and eastern part of northern China) during the last two decades. In fact, in most of the region, there is an increase in soil moisture, which has led to the reduction of dust emission (Figs. 2e and 2f). For the decline of surface winds, both anthropogenic global warming and internal climate variability can contribute to the decline, and there is no consensus on which factor is dominant currently. For soil wetting and surface greenness, both the anthropogenic global warming and internal climate variability can also contribute. Therefore, the decline of dust storm activity is a result of combined effects of three factors, and the effects are induced by both anthropogenic global warming and internal climate variability.

To avoid the confusion, we have revised the sentence and highlighted the main finding of this study in the Abstract: “Rather, the decline is part of the recent global climate change, driven by a combination of climatic factors but with the weakening of surface wind playing the dominant role”(Lines 32-33). We have also added more discussions

on the main findings and their implications in the final part of this manuscript: “On the other hand, global warming may induce soil moisture deficit and vegetation loss³⁵ and thus enhance dust activity, but this effect appears to be weak in the last two decades in Eastern Sources and may be overwhelmed by other factors” (Lines 246-248) and “Overall, this study highlights the combined effects of surface wind speed, soil moisture, and vegetation cover in the decline of East Asian dust activity in recent decades. Therefore, one needs to take into account the evolution of all these three factors to project the future change of dust activity in this region.” (Lines 266-270).

To specify the focus of the study, we also revised the title by adding “The roles of of climatic factors and ecological restoration”.

Line 55: “Activity”

Reply: Done.

Line 72: Add “of dust activity” after “decline”

Reply: Done.

Lines 210-211: About the wind regime, you may also see and discuss the westerlies over central Asia and the Tarim Basin, which highly modulate dust activity and loess formation over central Asia and NW China. In the respect, you may see a recent work by Li et al. (2021, Geoscience Frontiers) and many references therein. Changes in climatic factors i.e. pressure gradients, the Siberian High and Caspian-Sea High, may highly modulate the atmospheric circulation patterns, wind field and dust activity over the region.

Reply: We thank the reviewer for pointing out this. Following this suggestion, we have added more discussions on wind regimes. The dust storm activity in East Asia is greatly modulated by large-scale circulations including the Westerly Jet, the East Asia Trough, and the Siberian High (Wyrwoll et al., 2016; Li et al., 2021). Li et al. (2021) focused on the dust activity in Central Asia (to the west of Tarim Basin), which also helps us to understand the dust activity in Tarim Basin (western part of East Asian dust sources). We would like to mention that Caspian Sea High is not an important factor for East Asian dust activity to our knowledge, and we do not include it for discussion. In the revised manuscript, we have added the citations of Li et al. (2021) and other related references for the discussions on the wind regimes: “Strong surface winds for dust storms are associated with cold air outbreaks and cyclogenesis in the Mongolia Plateau^{9,24,28} and thus further influenced by the Westerly Jet, the East Asian Trough, and the Siberian High^{7,29}.” (Lines 230-232) and “Dust particles in Tarim Basin are mainly transported westward due to the blocking of surrounding mountains^{26,29}, while dust particles in Eastern Sources are mostly transported eastward and southward²⁰⁻²²” (Lines 545-548)

Lines 222-228: However, note that global warming lead to desertification, soil dryness and loss of vegetation which are positive feedbacks for increase in dust activity. This should not be ignored in the discussions here.

Reply: We thank the reviewer for pointing this out. We agree with the reviewer that global warming may lead to desertification, soil dryness and loss of vegetation, all of which can increase the dust activity. However, based on our analysis, this does not occur in the study region during the last two decades. In fact, in most of the regions, increase of soil moisture and vegetation is observed in this study (Figs. 2c, 2d, 2e and 2f), which contributes significantly to the decrease of dust emission in the region. To clarify, we have added the discussions on the positive feedbacks of global warming on dust emission: “On the other hand, global warming may induce soil moisture deficit and vegetation loss³⁵ and thus enhance dust activity, but this effect appears to be weak in the last two decades in Eastern Sources and may be overwhelmed by other factors” (Lines 246-248)

In the methods section, refer the period of the MERRA-2 retrievals.

Reply: Thank you for the comment. We have explicitly mentioned the period of the MERRA-2 retrievals and explained why we select 2001-2017 in this study: “MERRA-2 is provided from 1980 onwards and the data during 2001-2017 are used here. We select the period of 2001-2017 because the Moderate Resolution Imaging Spectroradiometer (MODIS) observation of leaf area index (LAI) is available from July 2000 and well-compiled synoptic records of dust storm are available before 2017, as will be described below. In addition, there are also advantages in focusing on the period of 2000s-2010s when much more observations are assimilated in MERRA-2 compared to the period of 1980s-1990s⁵⁶” (Lines 458-465).

Lines 417-418. The weather codes 06 and 07 also denote dust in suspension and blowing dust (see Hamzeh et al., 2021).

Reply: Thank you for the comment. We have added the information of weather codes in the description in Dust-in-Suspension and Blowing dust: “In addition to dust storms, two other kinds of dust events such as Dust-in-Suspension (DIS) and blowing dust (BD) are also recorded and denoted by the weather codes of 06 and 07, respectively. The definition is made by the World Meteorological Organization (WMO) and commonly used across the world^{2,63-64}” (Lines 529-532).

You should modify the colored scale in Fig. 2c, from green to blue colors.

Reply: Thank you for the suggestion. We have changed the colored scale from green to blue colors for positive values in Fig. 2c. Please find the new Fig. 2c in the revised manuscript.

Some recommended literature

- A. Emamian, A. Rashki, D.G. Kaskaoutis, A. Gholami, Ch. Opp, N. Middleton, 2021. Assessing vegetation restoration potential under different land uses and climatic classes in northeast Iran. Ecological Indicators, 122, 107325, <https://doi.org/10.1016/j.ecolind.2020.107325>*
- H. Gholami, A. Mohammadifar, H. Malakooti, Y. Esmaeilpour, S. Golzari, F. Mohammadi, Y. Li, Y. Song, D.G. Kaskaoutis, K.E. Fitzsimmons, A.L. Collins, 2021. Integrated modelling for mapping spatial sources of dust in central Asia - An important dust source in the global atmospheric system. Atmospheric Pollution Research 12, 101173, <https://doi.org/10.1016/j.apr.2021.101173>*
- Guan, Q., Yang, J., Zhao, S., (...), Zhang, D., Wu, T. 2015. Climatological analysis of dust storms in the area surrounding the Tengger Desert during 1960 – 2007. Clim. Dyn. Dyn.*
- M., Hamidianpour, S.M.A., Jahanshahi, D.G., Kaskaoutis, A., Rashki, P.G., Nastos, 2021. Climatology of the Sistan Levant wind: Atmospheric dynamics driving its onset, duration and withdrawal. Atmos. Res. 260,105711, <https://doi.org/10.1016/j.atmosres.2021.105711>*
- N.H. Hamzeh, D.G. Kaskaoutis, A. Rashki, K. Mohammadpour, 2021. Long-term variability of dust events in southwestern Iran and its relationship with the drought. Atmosphere, 12, 1350. <https://doi.org/10.3390/atmos12101350>.*
- S.K. Kharol, D.G. Kaskaoutis, K.V.S. Badarinath, A.R. Sharma, R.P. Singh, 2013. Influence of land use/land cover (LULC) changes on atmospheric dynamics over the arid region of Rajasthan state, India. Journal Arid Environments, 88, 90-101.*
- Y., Li, Y., Song, D. G. Kaskaoutis, J., Zan, R., Orozbaev, L., Tan, X., Chen, 2021. Aeolian dust dynamics in the Fergana Valley, Central Asia, since ~30 ka inferred from loess deposits. Geoscience Frontiers 101180, <https://doi.org/10.1016/j.gsf.2021.101180>.*
- Q., Luo, L., Zhen, Y., Xiao, H., Wang, 2020. The effects of different types of vegetation restoration on wind erosion prevention: a case study in Yanchi. Environ. Res. Lett. <https://doi.org/10.1088/1748-9326/abbaff>.*
- M., Tian, J., Gao, L., Zhang, H., Zhang, C., Feng, X., Jia, 2021. Effects of dust emissions from wind erosion of soil on ambient air quality. Atmos. Poll. Res. doi:10.1016/j.apr.2021.101108.*
- J., Xu, Y., Xiao, G., Xie, L., Zhen, Y., Wang, Y., Jiang, 2018. The Spatio-Temporal Disparities of Areas Benefitting from the Wind Erosion Prevention Service. Intern. J. Environ. Res. Public Health 15(7):1510, doi: 10.3390/ijerph15071510.*

Reply: We thank the reviewer for suggesting the relevant papers to us. In the revised manuscript, we have cited these papers and added the discussions on them.

References cited in the responses:

- Li, Y., Y. Song, D. G. Kaskaoutis, J. Zan, R. Orozbaev, L. Tan, and X. Chen (2021), Aeolian dust dynamics in the Fergana Valley, Central Asia, since ~30 ka inferred

from loess deposits, *Geoscience Frontiers*, 12(5), 101180, doi:<https://doi.org/10.1016/j.gsf.2021.101180>.

Randles, C. A., et al. (2017), The MERRA-2 Aerosol Reanalysis, 1980 Onward. Part I: System Description and Data Assimilation Evaluation, *Journal of Climate*, 30(17), 6823-6850, doi:10.1175/jcli-d-16-0609.1.

Wyrwoll, K.-H., J. Wei, Z. Lin, Y. Shao, and F. He (2016), Cold surges and dust events: Establishing the link between the East Asian Winter Monsoon and the Chinese loess record, *Quaternary Science Reviews*, 149, 102-108, doi:<https://doi.org/10.1016/j.quascirev.2016.04.015>.

Reviewer #3 (Remarks to the Author):

This study examined the trend in dust storm activity in China over 2001-2017, and used a physical model of dust emission to attribute such trend to different factors, including climatic and vegetation changes. The methods are sound and the results are scientifically valid, interesting and significant. There are remaining issues that should be addressed before this paper can be accepted for publication, as detailed below.

Reply: We thank the reviewer for a detailed review and helpful comments. The text, figures, and tables are revised as the reviewer suggested.

1. The overall scientific objective and methodology of this study are very similar to the one that was just recently published by Tai et al. (2021) (<https://doi.org/10.1016/j.atmosenv.2021.118348>). In some way this study is an improvement over that by Tai et al., which examined East Asian dust trend over 1980-2010, over which only about 10 years of actual observations are available, and most of their conclusions were based on modeled results. This study has examined nearly 20 years of more recent observations, and thus some of their conclusions were more solidly based on observations. Both studies have found climatic changes and variability to be the major factor shaping East Asian dust emission, while vegetation changes have played some minor roles. There are also some major differences, however. Tai et al.'s study has seen a recent reversal of wind trend and a corresponding increase in dust emission at least over 2000-2010 (which were also documented in the several studies cited by them). They also found that vegetation changes played a much smaller role in controlling dust emission overall than what this study has found. A more comprehensive discussion about the similarities and differences among the two studies is warranted, and a more thorough justification of the novelty of this work should be given.

Reply: We thank the reviewer for pointing out this very relevant study to us. It is of high interest to understand the underlying mechanisms of temporal variations in dust storm activity over East Asia. As pointed out by the reviewer, Tai et al. (2021) (hereafter Tai2021) have attempted to quantify the impacts of climatic factors and land cover change on the variations of East Asian dust storm activities during 1982-2010. Although our study shares some similarities in scientific objective and methodology with Tai2021, our study has advantages over Tai2021 in many aspects including methodology, model framework, and the model's ability to reproduce the historical change of dust activity, as listed in Table S3. Thus, our conclusions are more robust and comprehensive. The reasons are explained as follows.

- (a) First, our study adopted a more straightforward approach to quantify the impacts of various factors on dust emission. In contrast to Tai2021 that can only separate the total impacts of climatic factors from the impacts of vegetation cover, our study

takes a step forward to isolate the impacts from each of climate factors (surface wind, soil moisture, snow cover). To our knowledge, this is the first study that quantifies the impacts of each of climatic factors on the temporal variations of dust activity in East Asia. Our study presents a comprehensive assessment on the contributions of each of climate factors.

- (b) Second, our study developed an advanced modeling framework for dust emission in both model physics and input data. For model physics, although Tai2021 also uses a physically-based dust emission scheme as we do, Tai2021 only considers the effects of vegetation cover on threshold friction velocity and does not consider the variations of bare soil fraction due to vegetation cover changes by using a constant bare soil fraction (Kok et al., 2014). In contrast, the dust emission scheme in our study takes into account the impacts of vegetation cover changes on bare soil fraction (Eqs. 12 and 13). Therefore, Tai2021 underestimates the impacts of vegetation cover changes on dust emission.

For input data, the input land cover (LAI) data is based on fusion of MODIS (2000-2011) and AVHRR (1982-2000) in Tai2021, but this may introduce some inconsistency in the temporal variations of LAI between the two periods. By focusing only on the period after 2000, our study takes the advantage of MODIS that provides more accurate observations of LAI than AVHRR (Chen et al., 2019). In addition, our study also uses an improved reanalysis to force the model (MERRA-2 here compared to MERRA in Tai2021). The model is run at a higher horizontal resolution ($0.5^{\circ} \times 0.625^{\circ}$) here than that in Tai2021 ($2^{\circ} \times 2.5^{\circ}$).

- (c) Third, our model is evaluated in greater details and reproduces well the spatial-temporal variations of dust emission events, while Tai2021 lacks a detailed model evaluation against observations. As mentioned by the reviewer, the period for the model evaluation in Tai2021 is only 2000 to 2010 (excluding 2001), while their conclusions are mostly based on the comparison of three periods: 1980s, 1990s, and 2000s. Therefore, most of their conclusions were based on modeled results, as pointed out by the reviewer. In contrast, we evaluate the modeled dust emission in terms of spatial distribution, seasonal cycle, inter-annual variations, and long-term trends. The model agrees well with the observations in these aspects especially over Gobi Deserts (Figures 1a-1c, S3, S6, and S7). Particularly, our model simulates well the spatial distribution of dust emission regions in northern China and Tibetan Plateau (Figure S3), while Tai2021 simulates smaller dust emission regions (Figure S4 of Tai2021) compared to observations (Figure S3a). The inter-annual variations of simulated strong dust emission days and total dust emission flux in our model also agree very well with the observed dust storm days, with the correlations between simulations and observations being 0.78 ($p < 0.001$) and 0.73 ($p < 0.001$), respectively (Figures 1c and S7a).

In summary, our study is novel in many aspects including methodology, model

framework, and the model's ability to reproduce the observed dust activity, and thus provides a more reliable assessment of the impacts of various influencing factors (surface wind, soil moisture, and vegetation cover) on dust emission variations over East Asia during 2001-2017.

For “*the recent reversal of wind trend and a corresponding increase in dust emission at least over 2000-2010*” in Tai2021 mentioned by the reviewer, we do not see this change over Gobi Deserts according to the MISR observations (see Figure 4c of Tai2021). In fact, springtime average AOD over Gobi Deserts from MISR observations shows a slight decreasing trend during 2000-2010 (excluding 2001) (see Figure 4c of Tai2021). If 2001 is included in the analysis, the decreasing trend of AOD over Gobi Deserts during 2000-2010 is more pronounced as 2001 was the most dusty year during 2001-2017 (Figure 1c).

- (a) If the increasing trend of dust emission during 2000-2010 in Tai2021 mentioned by the reviewer is based on the model simulation there, we caution that the simulation results of Tai2021 show large discrepancy in both inter-annual variations and trends of AOD over dust source regions over East Asia. For inter-annual variations, the correlation coefficient between Tai2021 simulations and MISR AOD is around 0.1-0.43 during 2001-2010 (exclude 2001) over the Gobi Deserts and Taklimakan Deserts. For the trends, the model does simulate a slight increasing trend of AOD during 2001-2010 (excluding 2001), but this is opposite to a slight decreasing trend during 2000-2010 (excluding 2001) shown in observations. In addition, in Tai2021, we also caution that the increasing trend of simulated AOD is based on the years of 2000-2010 excluding 2001 (Figure 4c of Tai2021). If 2001 is included in the analysis, the increasing trends of simulated AOD is not present during 2000-2010 (according to Figure 7b of Tai2021).
- (b) We did see the increasing trend of dust emission over Tarim Basin during 2000-2010 (Figures 4b and 7c of Tai2021). This does not contradict the conclusion of our study as our study is focused on the Eastern Sources (not including Tarim Basin).
- (c) We did observe an increasing trend of dust activity in 2000-2002 compared to previous decades (i.e., 1990s) (see our previous study such as Wu et al. (2018)). However, this increasing trend during 1990s and early 2000s is beyond the scope of this study, as our study focuses only on the last two decades when more accurate observations of LAI from satellite measurements are available.
- (d) We went over the references cited by Tai2021, and however, we did not find any of these studies mentioning the increasing trends of dust emission/storm activity during 2000-2010 over East Asia. Instead, the statement of an increase trend in dust emission/storm activity in some references cited by Tai2021 refers to the comparison of 2000s to 1990s. It is beyond our scope to compare early 2000s and 1990s.

For “*the much smaller role of vegetation cover in controlling dust emission overall in Tai2021 than this study*” mentioned by the reviewer, we would like to note that our study provides a more reliable estimation. We demonstrate that 30% of dust emission reduction in 2010-2017 compared to 2001 can be attributed to vegetation cover change. As we replied to the reviewer’s comment above, the smaller role of vegetation cover in Tai2021 than our study may be attributed to various aspects, particularly the model physics and input data. For the model physics, Tai2021 does not take into account the impacts of vegetation cover change on dust emission (i.e., Tai2021 uses a constant bare soil fraction map for dust emission), and thus Tai2021 underestimates the impacts of vegetation cover change on dust emission. For input data, our study uses the LAI dataset solely based on the MODIS observations, while Tai2021 used the fusions of two LAI datasets (AVHRR and MODIS). Previous study of Chen et al. (2019) has shown that the increasing trend of LAI is more pronounced in the MODIS dataset than the AVHRR dataset over the same period. The differences in dust emission parameterization and input data mainly explain the difference in the impacts of vegetation cover change between Tai2021 and this study. Other factors such as meteorological forcing and model resolution may also contribute to the difference, but it is difficult for us to trace them. Overall, despite the difference in the impacts of vegetation cover change in Tai2021 and our study, this does not induce any contradiction and our results are more reliable.

In the revised manuscript, we have cited Tai2021 and added a comprehensive discussion about the similarities and differences between our study and Tai2021 to justify the novelty of our study:

Lines 65-73: For a quantitative assessment, it is vital to rely on numerical models as they can be used to study dust activity over an entire region and to quantify the contributions from individual factors. A recent study has attempted to quantify the impacts of climatic factors and vegetation cover on dust emission variations over East Asia during 1982-2010, but the results remain questionable as the model they used is not fully validated¹⁹. Compared to the observations of aerosol optical depth in 2000-2010 excluding 2001, the model shows low fidelity in the temporal variations of dust activity¹⁹. Limitations in that study call for a more reliable quantitative assessment with an improved model that can reproduce dust variations shown in the observations.

Lines 77-79: The model is validated with dust storm records (Figure S2) in terms of spatial and temporal variations and thus provides a reliable estimation of dust emissions.

Lines 490-496: Although a recent study has adopted a similar approach to isolate the impacts of climatic factors and vegetation cover changes¹⁹, our study has great advantages over it in terms of the modeling framework, including dust emission parameterizations to account for the impacts of vegetation cover, meteorology forcing, and input data of LAI (Text S2). Our study also makes a step forward to

separate the impacts from soil moisture and surface wind, while the impacts of these two factors are not separated but considered together in that study.

Supplementary information

Text S2

Comparison of this study with a recent study of Tai et al. (2021).

This study shares some similarities with a recent study¹⁹ (hereafter Tai2021), which also set up a modeling framework to separate the impacts of climatic factors and vegetation cover change on the variations of dust storm activity in East Asia. The comparison of this study with Tai2021 is given in Table S3. Overall, although with the similar approach, this study has advantages over Tai2021 in several key aspects, including dust emission parameterizations to account for the impacts of vegetation cover, improved meteorology forcing, more accurate LAI dataset, and higher spatial resolution. Therefore, the results in this study are more reliable than those of Tai2021.

Table S3. Comparison in key aspects between this study and Tai2021

Aspects	This study	Tai2021
Main focus		
Objective	Isolate the impacts of surface wind speed, soil moisture, and vegetation cover on the variations of dust storm days.	Isolate the impacts of climatic factors (combine surface wind, precipitation, temperature) and vegetation cover on the variations of dust storm days.
Period	2001-2017	1982-2010
Methodology		
Sensitive experiments	Impacts of surface wind speed, vegetation cover, and soil moisture are separated with each factor fixed in 2001.	Impacts of climatic factors (including surface wind speed, precipitation, and temperature) are combined together and separated from those of land cover change with meteorology or land cover fixed in 1995.
Modeling framework		
Model	A physically-based dust emission model (DuEM v.10)	A chemical transport model (GEOS-Chem) with a physically-based dust emission module ⁷⁰
Parameterizations on the impacts of	The impacts of vegetation cover change on both bare	Only the impacts of vegetation cover change on threshold

vegetation cover on dust emission	soil fraction (Eqs. 12-13) and threshold friction velocity are considered (Eqs. 9-10)	friction velocity are considered (Eq.3 of Tai2021)
Meteorological forcing	MERRA-2 ⁵⁶	MERRA ⁷¹
LAI data*	MODIS ^{60,61}	Fusion of MODIS (2000-2011) and AVHRR (1982-2010) ⁷²
Horizontal resolution	0.5° × 0.625°	2° × 2.5°
Results		
Model performance	The model performs well in the spatial distribution of dust emission regions, and seasonal cycle, interannual variations and trends of dust storm days in East Asia especially in Eastern Sources.	The simulated dust optical depth is compared to observations during 2000-2010 (excluding 2001), and the model performs poorly in the trends of AOD over Taklimakan Deserts and Gobi Deserts during 2000-2010 (excluding 2001).

*: Better quality LAI is obtained by MODIS than by AVHRR¹⁵.

2. One important difference of this study relative to Tai et al. (2021) is that the model this study used only simulated dust emission, but not the subsequent transport, deposition and thus concentration as did Tai et al. (2021). This makes comparison with observations harder because most of the long-term observations are concentrations or aerosol optical depths, which a dust emission model alone cannot give. This study did compare their simulated dust emission with that provided by MERRA-2; yet, the dust emission product of MERRA-2 is not actual observations, but another model output from their assimilation that also made use of model schemes that were similar to what this study used. Therefore, the model-observation comparison of this study appears to be not entirely based on independent observations; these need to be discussed and addressed in the paper.

Reply: We thank the reviewer for the comment. In this study, we focus on understanding the variations of dust emission flux in East Asia, as dust emission is first and foremost component of dust cycle. We agree with the reviewer that “***One important difference of this study relative to Tai et al. (2021) is that the model this study used only simulated dust emission, but not the subsequent transport, deposition and thus concentration as did Tai et al. (2021).***”, but for dust emission we focused on, we did use the independent observations to validate the model performance. The observations we used are based on the dust event records at synoptic stations distributed in northern China (Please see Methods: (4) Dust storm records at synoptic stations), which are independent of the model simulation.

The synoptic stations are widely distributed in northern China (Figure S2). The synoptic observations consist of continuous records of dust activity in last decades. The dataset is generated after the strict quality control by the China Meteorological Administration and shows high quality. The synoptic records of dust storms are widely used in previous studies to characterize the spatial-temporal variations of dust activities in East Asia (e.g., Shao, 2008; Guan et al., 2015, 2017; Wyrwoll et al., 2016; Wu et al., 2018; Wang et al., 2021). Although dust storm records cannot be directly used to estimate dust emission, it is generally associated with strong dust emission. Therefore, we compare the simulated strong dust emission days with observed dust storm days (Figure 1c). The results show the model captures well the temporal variations of dust storm days in East Asia during 2001-2017 (with $R=0.78$, $p<0.001$).

In addition, we would like to clarify that dust emission flux in MERRA-2 is not used to validate our dust emission model. As mentioned by the reviewer, dust emission flux of MERRA-2 is not actual observations, but another model output by dust emission scheme that was similar to what this study used. We compare our simulation with dust emission flux of MERRA-2 to demonstrate the better performance of our model, by comparing both of them (our model and MERRA-2) to real observations (i.e., synoptic observations of dust storm days which are mentioned in the paragraph above). As shown in Figures S3, S6, S16, and S17, compared to synoptic observations, our model shows better performance in simulating the spatial-temporal variations of dust emission flux over East Asia than MERRA-2. We have clarified this in Lines 498-501: “Note that although MERRA-2 official dataset also provides the hourly dust emission flux along with meteorological fields, dust emission flux generated by our model agrees better with the observations than that from MERRA-2 official dataset (Figures S3, S6, S16, and S17).” and Lines 506-510: “Our model not only simulates dust emission over deserts, but also over other sparsely-vegetated lands with bare soils such as grasslands and croplands (Figures S14 and S15) where there are significant seasonal variations in vegetation cover (Figures S18 and S19). Therefore, our model reproduces better the spatial distributions and seasonal variations of dust emission in East Asia than MERRA-2 official dataset.”.

On the other hand, although this study focuses on the dust emission, we agree that inclusion of dust transport and deposition is important for understanding the full dust cycle. Therefore, to take the advantage of long-term observations of aerosol optical depth, as pointed out by the reviewer, we also include the dust aerosol optical depth (DAOD) derived from both MODIS and CALIOP (Song et al., 2021), to demonstrate the decreasing trend of dust activity in both dust sources regions and downwind regions over East Asia (Figures S4 and S5). The temporal variations of dust emission flux are also compared with the DAOD over Eastern Sources and the results show that the model simulates well the interannual variations of dust activity in this region with the significant correlation ($R=0.58$ and $R=0.66$ for MODIS/Aqua and CALIOP, respectively; $p<0.05$) between simulated dust emission and observed DAOD (Figure S7a). Therefore, although our study cannot directly simulate dust transport, these results

reinforce one key conclusion of our study that the simulated dust emission reflects well the temporal variations of dust activity during the last two decades over East Asia.

In the revised manuscript, we have included the decreasing trend of satellite DAOD “The decreasing trend of dust activity is also reflected in satellite measurements of dust aerosol optical depth (DAOD) in both the dust source regions and downwind regions²³⁻²⁴ (also see Figures S4 and S5)” (Lines 107-110) and its comparison with simulated dust emission over Eastern Sources: “The simulated regionally-accumulated dust emission amount is also well correlated with the regional mean dust storm days ($R=0.73$, $p<0.001$) and DAOD ($p<0.05$), as shown in Figure S7a”(Lines 142-144). We have also added a discussion on the inclusion of dust transport for assessing the benefits of reduced dust emission: “On the other hand, due to the impacts of dust transport and deposition, the benefits of reduced dust emission in alleviating dust pollution may vary in different regions and should be assessed with full consideration of dust transport¹⁹⁻²¹” (Lines 263-266).

In summary, our model simulation is compared to synoptic observations that are independent of our simulation, and the results demonstrate the reliability of our model in reproducing the spatial-temporal variations of dust emission flux over East Asia during 2001-2017. Our simulation of dust emission flux also performs better than that from MERRA-2 official dataset. Following the reviewer’s suggestion on aerosol optical depth, we also include the dust aerosol optical depth (DAOD) from satellite observations to demonstrate the decreasing trend of dust activity and evaluate the model simulated dust emission flux over Eastern Sources. Therefore, the model-observation comparison of this study is entirely based on independent observations, and the consequent conclusions are solid and robust.

3. The authors linked the weakening wind trend to anthropogenic activities that contribute to global warming, and thus related the dust trend to anthropogenic activities in the broader sense. While the general changes in midlatitude wind over large areas can be attributable to global warming, as many studies have found, specifically attributing the past wind changes over the desert and semiarid regions of concern in this study to anthropogenic activities at large requires a lot more climate model experiments in the form of attribution studies than this study has afforded to do. Therefore, the linkage of their changes found, which are solid, to anthropogenic activities as they claimed, is tenuous at best. Therefore, either the authors should tone down this claim, or discuss the connection with much greater details and nuisances.

Reply: We thank the reviewer for the comment. We would like to mention that although not specifically focusing on the dust source regions in East Asia, previous studies did show the weakening of surface wind speed over dust source regions of East Asia due to global warming in the figures they have presented (e.g., Figure 3 of Karnauskas et al., 2018; Figures 2, 3, 7 of Shen et al., 2021). The gap in the attribution of past wind change for the dust source regions of East Asia is to what

extent the weakening of surface winds can be ascribed to global warming as the decadal climate variabilities (e.g., Pacific Decadal Oscillation) may also play a role. In this respect, we do agree with the reviewer that the attribution of weakening wind trend over the dust source regions to different factors (natural variability and global warming) is complicated, which needs more work on the attribution of past wind change and is beyond the scope of this study.

Therefore, in the revised manuscript, we have clarified the statement “**The weakening of surface wind speed in the mid-latitudes of North Hemisphere can be ascribed to the reduced meridional temperature gradients due to the polar amplification of global warming^{25,26}**” by adding “**although the magnitude of the effect over the dust source regions is unclear**” after it (Lines 242-243). We have also removed the statement “**With projected increases in GHG concentrations and global warming in the near future²⁷, the dust storm activities may continue to decrease in a warming climate.**” in the revised manuscript.

We have also de-emphasized the role of global warming in past wind changes, highlighted the main findings of this study related to the combined effects of three factors on dust activity, and discussed their implications in the final part of the manuscript: “**Overall, this study highlights the combined effects of surface wind speed, soil moisture, and vegetation cover in the decline of East Asian dust activity in recent decades. Therefore, one needs to take into account the evolution of all these three factors to project the future change of dust activity in this region.**” (Lines 266-270)

4. The article is at times too colloquial and informal for a scientific paper, e.g., the use of apostrophes and short forms. These should be avoided, and the article should be rewritten in a more formal tone.

Reply: We thank the reviewer for the comment. We have carefully read through the manuscript and made substantial changes in the language. Some revisions are listed below:

- Delete “Dust storms have a long history in China”;
- Change “is down to” to “can be attributed to” (Line 29);
- Change “doesn’t ”to “does not” (Lines 30-31);
- Change “the battle against anthropogenic air pollution” to “combatting the anthropogenic air pollution” (Line 40);
- Change “unknown” to “unclear” (Line 41);
- Change “is going to” to “will” (Line 41);
- Change “a determining critical role” to “a critical role” (Line 55);
- Change “determine the contributions” to “quantify the contributions” (Line 64);
- Change “agree well with” to “is consistent with” (Lines 119-120);
- Change “Dust emission is not changed equally for each month” to “Changes of dust emission show large seasonal variations” (Line 193);
- Change “explained by” to “attributed to” (Line 226);

Delete “which people undoubtedly benefit from”;

For the short forms of expressions, we used them in the tables and figures for easy reading. Following the reviewer’ suggestion, we have further checked the captions for tables and figures and added the full explanations of these forms.

Other changes can also be found throughout the manuscript. We believe the changes have greatly improved the manuscript and made it a scientific paper for the journal.

References cited in the responses:

- Chen, C., et al. (2019), China and India lead in greening of the world through land-use management, *Nature Sustainability*, 2(2), 122-129, doi:10.1038/s41893-019-0220-7.
- Guan, Q., X. Sun, J. Yang, B. Pan, S. Zhao, and L. Wang (2017), Dust Storms in Northern China: Long-Term Spatiotemporal Characteristics and Climate Controls, *J Climate*, 30(17), 6683-6700, doi:10.1175/jcli-d-16-0795.1.
- Guan, Q., J. Yang, S. Zhao, B. Pan, C. Liu, D. Zhang, and T. Wu (2015), Climatological analysis of dust storms in the area surrounding the Tengger Desert during 1960–2007, *Climate Dynamics*, 45(3), 903-913, doi:10.1007/s00382-014-2321-3.
- Karnauskas, K. B., J. K. Lundquist, and L. Zhang (2018), Southward shift of the global wind energy resource under high carbon dioxide emissions, *Nat Geosci*, 11(1), 38-43, doi:10.1038/s41561-017-0029-9.
- Kok, J. F., et al. (2014), An improved dust emission model – Part 1: Model description and comparison against measurements, *Atmos. Chem. Phys.*, 14(23), 13023-13041, doi:10.5194/acp-14-13023-2014.
- Shao, Y. (2008), *Physics and Modelling of Wind Erosion*, 452 pp., Springer, Berlin, Germany.
- Shen, C., J. Zha, D. Zhao, J. Wu, W. Fan, M. Yang, and Z. Li (2021), Estimating centennial-scale changes in global terrestrial near-surface wind speed based on CMIP6 GCMs, *Environmental Research Letters*, 16(8), 084039, doi:10.1088/1748-9326/ac1378.
- Song, Q., Z. Zhang, H. Yu, P. Ginoux, and J. Shen (2021), Global dust optical depth climatology derived from CALIOP and MODIS aerosol retrievals on decadal timescales: regional and interannual variability, *Atmos. Chem. Phys.*, 21(17), 13369-13395, doi:10.5194/acp-21-13369-2021.
- Tai, A. P. K., P. H. L. Ma, Y.-C. Chan, M.-K. Chow, D. A. Ridley, and J. F. Kok (2021), Impacts of climate and land cover variability and trends on springtime East Asian dust emission over 1982–2010: A modeling study, *Atmos Environ*, 254, 118348, doi:10.1016/j.atmosenv.2021.118348.
- Wang, S., Y. Yu, X.-X. Zhang, H. Lu, X.-Y. Zhang, and Z. Xu (2021), Weakened dust activity over China and Mongolia from 2001 to 2020 associated with climate change and land-use management, *Environmental Research Letters*, 16(12), 124056, doi:10.1088/1748-9326/ac3b79.

- Wu, C., Z. Lin, X. Liu, Y. Li, Z. Lu, and M. Wu (2018), Can Climate Models Reproduce the Decadal Change of Dust Aerosol in East Asia?, *Geophys Res Lett*, 45(18), 9953-9962, doi:10.1029/2018gl079376.
- Wyrwoll, K.-H., J. Wei, Z. Lin, Y. Shao, and F. He (2016), Cold surges and dust events: Establishing the link between the East Asian Winter Monsoon and the Chinese loess record, *Quaternary Science Reviews*, 149, 102-108, doi:<https://doi.org/10.1016/j.quascirev.2016.04.015>.

REVIEWER COMMENTS

Reviewer #1 (Remarks to the Author):

We have previously reviewed this manuscript "Drivers of recent 1 decline in dust activity over East Asia: The roles of climatic factors and ecological restoration". However, the author did not adopt our comments and suggestions, I would suggest the authors address further the following aspects, I read this manuscript carefully and found that there are still some questions, listed below:

1. In the abstract, the author mentions that the weakening of near-surface wind and increases in vegetation cover and soil wetness have all contributed to the recent decline in dust storm activity. The authors then mention that the three drivers contribute 46%, 30 and 24% respectively, which sums to 100%, which makes me confused. Are these three factors alone contributing to the decrease in dust storm activity? This is not the right proportion when the other factors are not fully considered, so I think the author's expression here is very inappropriate.
2. The innovations in this manuscript are not well clarified by the author.
3. why does the author focus only on wind speed, vegetation cover and soil moisture? Why are other factors such as temperature, precipitation, human activity, etc. not considered?
4. Lines 80-83, There is less than one dust storm day per year on average in the other regions of northern China (north of 35°N)? Please check!
5. The description and order of the figures in the results section is rather confusing and should have been adjusted by the author as appropriate.
6. What is the author's basis for treating regional accumulated dust emission > 4 Tg day⁻¹ for the entire eastern source area as a strong dust emission day? Furthermore, a strong dust emission day is the sum of emissions from all stations, so it is not reasonable to compare a strong dust emission day for the whole region with the regional average dust days.
7. Lines 131 - Drivers of weakening dust storm activity in last two decades : I remain rather confused by the choice of drivers of dust storms in the manuscript. The authors do not explain why wind speed, leaf area index, soil moisture, and snow cover were used as drivers of dust storm activity? In addition, the rest of the natural factors such as temperature and precipitation are completely ignored compared to snow cover, why?
8. Lines 131 - Drivers of weakening dust storm activity in last two decades : Dust storms are natural processes due to multiple factors and the effect of a single factor on dust emission fluxes alone is not sufficient to describe the drivers of dust storm activity. Multi-factor integrated analysis models or methods should be used. For example, the southern part of the Qinghai-Tibet Plateau, where wind speeds are high, has relatively low dust emissions.
9. The analysis of vegetation trends in Figures 2c and 2d is not very accurate, and the colors of the figure legends do not seem to match.
10. The authors only analyzed vegetation, wind speed and soil moisture, and the contribution of all three to dust emissions was 100%, which is not very reasonable. Furthermore, I believe that dust emissions are a complex process and that the results obtained from single factor analysis are not sufficient to describe the main drivers of dust emissions and their contribution, therefore, a multi-factor integrated analysis would be indispensable indeed.
11. How were the variation curves for the factors in Fig. 3 obtained?
12. The authors obtained that wind speed is the main contributor to dust storm emissions and that April is the season with the largest number of strong dust storm days and dust storm days (Fig. S4), but how is it understood that in Fig. S7, dust emission fluxes are significantly higher in autumn than in spring and summer? In addition, vegetation cover is relatively better in autumn than in winter and spring, but dust emissions are higher than in other seasons? How can this logic be understood?
13. Lines 175-191, The authors mention that soil wetness is caused by precipitation and evaporation, but in summer evaporation is generally relatively strong in semi-arid areas, so why is the soil moister in summer than in other seasons? Vegetation is better in the autumn; wind speeds are higher and dust emissions are most variable in the autumn. This confuses me, the logic between vegetation and wind speed is not clearly stated and although wind speed plays a greater role than vegetation, it is difficult to convince the

reader of a single factor alone.

14. The authors should have given a discussion and analysis of the seasonal variation in drivers and the relationship between drivers and dust emissions, rather than simply describing the phenomenon.

15. Lines 203-204: Only 23% of dust emission reduction from 2003 2001 to the 2010s in grasslands were a result of vegetation greening. Where do I get it from?

Reviewer #3 (Remarks to the Author):

I thank the authors for their extensive revision and responses to my previous comments. My original comments regarding the comparison with Tai et al. (2021) were intended to ensure the authors would acknowledge all past, especially the recent, work that examined the same issue, giving credits to them and carefully explaining the new aspects, possible improvements and new insights offered by their own work in the context of previous work. They were not asked to be the judge to give a definitive statement on which is "better", which would be an unfair conclusion either way due exactly to the differences in assumptions and methodologies employed. However, the authors have tried extensively to discredit the previous work and for many times stated that their work was far "better", exaggerating the novelty of their own work that is simply not grounded in their methods, results and conclusions. Such an overcritical approach also undermines the scientific objectivity and professionalism of their manuscript. Therefore, I do not recommend the publication of this manuscript unless the authors can revise the manuscript, taking a more objective and comparative approach, giving full credit to and acknowledging the significance of previous work, and simply highlighting what they have done differently and how their work offers new insights that the previous work may not convey.

More specifically:

"A recent study has attempted to quantify the impacts of climatic factors and vegetation cover on dust emission variations over East Asia during 1982-2010, but the results remain questionable as the model they used is not fully validated. Compared to the observations of aerosol optical depth in 2000- 2010 excluding 2001, the model shows low fidelity in the temporal variations of dust activity."

- The model the previous work used was adequately validated using multiple datasets that are more varied and comprehensive than the authors' work. For instance, this current study only used dust emission observations, which are extremely sparse, as well as dust storms, which are more comprehensive, whereas the previous work used a variety of data for dust storms, concentrations and optical depths. This is not a limitation of this work, though, because this work focuses only on dust emissions while the previous work focused on the whole atmospheric cycle of dust. Due to the different scopes of the studies, it is unnecessary and rather irrelevant to erroneously discredit the previous work to "call for a more reliable quantitative assessment with an improved model". A due discussion of the previous work, highlighting the key findings and significance, while stating the different aspects and perspectives that their own work can bring, is enough.

"The model is validated with dust storm records (Figure S2) in terms of spatial and temporal variations and thus provides a reliable estimation of dust emissions."

- There is no guarantee that dust storm activities provide more "reliable" evidential basis than other dust parameters such as dust optical depths or concentrations. This statement on its own is simply not necessary.

"Our study has great advantages over it in terms of the modeling framework, including dust emission parameterizations to account for the impacts of vegetation cover, meteorology forcing, and input data of LAI (Text S2)"

- Again, the authors were not asked to be the judge, and indeed those "great advantages" claimed are rather dubious. I fully acknowledge that the emission model of dust per se has several aspects that more comprehensively represent the physical dependence of dust emissions on various factors. But comprehensiveness of complexity of model processes do

not automatically lend the model greater credence, especially without long-term observations of the direct physical outcomes to validate (i.e., dust emission flux). Also, the authors fail to acknowledge that a key aspect of Tai et al. (2021) that their study did not address is the atmospheric transport and deposition of dust as simulated in a full chemical transport model, which for sure introduce more uncertainties due to the larger scope involved but can then make use of a larger variety of datasets to validate model outcomes. Such key outcomes and the major conclusions therein were not discussed at all by the authors, who have opted to focus on unfairly discrediting the previous work.

"The results in this study are more reliable than those of Tai2021"

- This is simply not grounded by evidence, and is indeed an unnecessary comparison to make to begin with. As stated by the authors themselves, the long-term observations used that are the most comprehensive are dust storm days, but the physical connections between dust emission per se and dust storm activity are far from immediate - dust storm activity can be further modulated by other processes such as transport and deposition, which were not considered by the authors' model but considered by Tai et al. (2021). Understandably, different models have different functions, aims and uncertainties, and saying one is more reliable than other, especially in this case, is like comparing oranges to apples.

"On the other hand, due to the impacts of dust transport and deposition, the benefits of reduced dust emission in alleviating dust pollution may vary in different regions and should be assessed with full consideration of dust transport."

- This is exactly an "advantage" of Tai et al. (2021), but the authors fail to acknowledge this.

REVIEWER COMMENTS

Reviewer #1 (Remarks to the Author):

We have previously reviewed this manuscript “Drivers of recent decline in dust activity over East Asia: The roles of climatic factors and ecological restoration”. However, the author did not adopt our comments and suggestions, I would suggest the authors address further the following aspects, I read this manuscript carefully and found that there are still some questions, listed below:

Reply: We thank the reviewer for his/her constructive reviews and comments. The reason why we did not adopt your comments in last round is that your reviews were not forwarded to us by the editor. We are very sorry for that. We really appreciate your reviews in this round on our manuscript. We have carefully considered your comments and revised the text, tables, and figures following your suggestions.

1. In the abstract, the author mentions that the weakening of near-surface wind and increases in vegetation cover and soil wetness have all contributed to the recent decline in dust storm activity. The authors then mention that the three drivers contribute 46%, 30 and 24% respectively, which sums to 100%, which makes me confused. Are these three factors alone contributing to the decrease in dust storm activity? This is not the right proportion when the other factors are not fully considered, so I think the author's expression here is very inappropriate.

Reply: We thank the reviewer for pointing out this. We would like to mention that we quantify the impacts of surface wind speed, soil moisture, and vegetation cover as they are the three main factors which can directly affect the dust emission. The impact of another factor, snow cover, is also examined, and the result show that snow cover has some impacts on the seasonal variations of dust emission but has little impacts on the trends of dust emission during 2001-2017 over East Asia. All together the three factors (surface wind speed, soil moisture, and vegetation cover) can explain mostly of the interannual variability and trends in dust emission simulated by our model. Some other

climatic factors such as temperature and precipitation can affect the dust emission indirectly through their impacts on the aforementioned factors.

For the impacts of human activities, we consider the ecological restoration as a major type of human activities that can affect the dust emission in large areas. It affects the dust emission indirectly by changing the vegetation cover, which is considered in our model. In addition to ecological restoration, other human activities such as the cultivation and grazing can perturb the soils and change the soil cohesion, but it is difficult to quantify these impacts in large areas as our knowledge on these processes is limited and our dust emission model does not consider these impacts. However, their impacts may be much smaller compared to those of ecological restoration, as the ecological restoration can change the land surface cover in large areas.

Following the reviewer's comment, we have revised our statements on the values of contributions. First, we explain the contribution ratios (i.e., 46%, 30%, 24%) as “**the relative contributions**” among the three factors themselves in Abstract (Line 19). Second, we add “**mostly**” when highlighting the role of climatic factors in driving the recent decline of dust activity (Line 24).

We also clearly explain the assumptions in our model: “**In the model, the impacts of surface wind speed, soil moisture, vegetation cover, and snow cover on the dust emission are explicitly represented (see Methods). Other factors (e.g., temperature and precipitation) can indirectly impact the dust emission through affecting these factors.**” (Lines 83-86).

For the impacts of ecological restoration and other human activities, we have added a discussion: “**In this study, we consider the ecological restoration as a major type of human activities that can affect the dust emission in large areas. Our model results show only a minor contribution from the ecological restoration to the recent decline of dust activity in East Asia. In addition to the ecological restoration, other human activities**

may perturb soils directly through cultivation and grazing, which is not considered in the model. However, the perturbations generally lead to enhanced dust emission by reducing soil cohesion, which may cancel out some of the dust emission reduction due to the ecological restoration.” (Lines 267-274).

To summarize, we have changed “the contributions from three factors” to “**the relative contributions of these three factors**”. We have explained why we focus on the three key factors, and also discussed other factors which are not considered in the model. Despite these changes, the main conclusions of our study remain unchanged.

2. The innovations in this manuscript are not well clarified by the author.

Reply: We thank the reviewer for the comment. We have made the innovations of this study clearer in the revised manuscript. The major innovation is that we quantify the impacts of surface wind speed, soil moisture, and vegetation cover, which may have important implications for understanding the historical dust variations and projecting its future changes. Although the impacts of climatic factors and vegetation cover have been quantified before, the previous studies focused on the decades of 1980s to 2006/2010, while our study focuses on the most recent two decades when vegetation cover in East Asia due to ecological restoration is more intensive than the earlier decades. Another innovation of this study is that we have made a step forward in methodology by separating the impacts of surface wind speed and soil moisture, while previous studies combined their impacts together.

In the revised manuscript, we have clarified our innovations in the Abstract: “**It is essential to understand the factors driving the recent decline of dust activity in East Asia for projecting the future dust activity.**” (Lines 15-16), in the Introduction: “**These two studies focused on the period of 1982 to 2006/2010, and it is still unclear about the roles of climatic factors and vegetation cover in the last two decades when more intensive ecological restorations were implemented¹⁵**” (Lines 75-78), and in the Methods: “A

recent study has adopted a similar approach to isolate the impacts of climatic factors and vegetation cover²¹. In that study, the impacts of soil moisture and surface wind were considered together. Our study makes a step forward by separating the impacts of these two factors.” (Lines 549-552).

3. why does the author focus only on wind speed, vegetation cover and soil moisture? Why are other factors such as temperature, precipitation, human activity, etc. not considered?

Reply: We thank the reviewer for the questions. In this study, we focus on the dust emission, which is the first and foremost component of dust cycle. We quantify the impacts of surface wind speed, soil moisture, vegetation cover, and snow cover as they are the factors which can directly affect the dust emission. In addition, other climatic factors such as temperature and precipitation can affect the dust emission indirectly through their impacts on the four factors we have examined. We have clarified this in the revised manuscript: “In the model, the impacts of surface wind speed, soil moisture, vegetation cover, and snow cover on the dust emission are explicitly represented (see Methods). Other factors (e.g., temperature and precipitation) can indirectly impact the dust emission through affecting these factors.” (Lines 83-86).

For the human activities, indeed we do not consider them directly, but the impact of the ecological restoration, which is a major type of human activities that can affect the dust emission, has been considered through its modifications on the vegetation cover in our study. In addition to the ecological restoration, other human activities such as cultivation and grazing can affect the dust emission directly by modifying the soil cohesion, which are however not considered in our study due to the limited knowledge of relevant processes.

In the revised manuscript, we have clarified the impacts of human activities on the dust emission: “In this study, we consider the ecological restoration as a major type of human

activities that can affect the dust emission in large areas. Our model results show only a minor contribution from the ecological restoration to the recent decline of dust activity in East Asia. In addition to the ecological restoration, other human activities may perturb soils directly through cultivation and grazing, which is not considered in the model. However, the perturbations generally lead to enhanced dust emission by reducing soil cohesion, which may cancel out some of the dust emission reduction due to the ecological restoration.” (Lines 267-274).

4. Lines 80-83, There is less than one dust storm day per year on average in the other regions of northern China (north of 35°N)? Please check!

Reply: We thank the reviewer for pointing out this. We have clarified the description by changing “in the other regions of northern China (north of 35°N)” to “**to the east and the south of the deserts/sandy lands**” (Line 102).

5. The description and order of the figures in the results section is rather confusing and should have been adjusted by the author as appropriate.

Reply: We thank the reviewer for pointing out this. We have adjusted the order of the figures and associated text and described the figures in order, such as Figure 1, Figure 2, Figure 3, ...

6. What is the author's basis for treating regional accumulated dust emission $> 4 \text{ Tg day}^{-1}$ for the entire eastern source area as a strong dust emission day? Furthermore, a strong dust emission day is the sum of emissions from all stations, so it is not reasonable to compare a strong dust emission day for the whole region with the regional average dust days.

Reply: We thank the reviewer for the comment. The basis for selecting regional accumulated dust emission $> 4 \text{ Tg day}^{-1}$ over Eastern Sources as a criterion for a strong

dust emission day is that the regionally-mean dust emission flux is greater than $10 \mu\text{g m}^{-2} \text{s}^{-1}$ over this criterion (i.e., $> 4 \text{ Tg day}^{-1}$). $10 \mu\text{g m}^{-2} \text{s}^{-1}$ is based on the field observations of dust emission flux (Shao et al., 2011), which showed the threshold dust emission flux (dust particles with the diameters $< 8.4 \mu\text{m}$) for a dust emission event is between 1 to $10 \mu\text{g m}^{-2} \text{s}^{-1}$. Therefore, for dust particles with diameters $< 20 \mu\text{m}$, we assume that the threshold dust emission flux is around $10 \mu\text{g m}^{-2} \text{s}^{-1}$.

We agree with the reviewer's comment "*it is not reasonable to compare a strong dust emission day for the whole region with the regional average dust days*" as the observational stations are only located in part of the region. To eliminate the mismatch in the comparison, we have removed the strong dust emission days but calculated the mean dust emission flux at the observational stations. The results are shown in Figure 1e and S5a. From Figure 1e, we can see that the model generally simulates well the interannual variations and trends of dust activity during 2001-2017 over East Asia. From Figure S5a, we can see that the model simulates reasonably well the seasonal variations of dust activity, although the model overestimates the dust emission in the late autumn and winter, which is partly due to the underestimation of snow cover in the model (Reichle et al., 2017a).

In addition, it is also desirable to derive the number of dust emission days from the model simulations so as to evaluate the model's performance. Therefore, we also calculate the number of dust emission days from the model simulations for each of the observational stations and average them to derive the regionally-mean dust emission days for all the stations, as done in the observations. To consider the uncertainty in the selection of the threshold value for a dust emission day, we use three criteria: $> 1 \mu\text{g m}^{-2} \text{s}^{-1}$, $> 10 \mu\text{g m}^{-2} \text{s}^{-1}$, and $> 50 \mu\text{g m}^{-2} \text{s}^{-1}$, respectively. The comparison of model simulated regionally-mean dust emission days with observed regionally-mean dust storm days are shown in Figures S5b-S5d and S8. Overall, the comparison results are similar to those shown in Figures S5a and 1e.

In the revised manuscript, we have clearly described the comparison results and the basis for the selection of the threshold values: “The simulated mean dust emission fluxes at the synoptic stations decrease significantly ($p < 0.1$) from $0.75 \mu\text{g m}^{-2} \text{s}^{-1}$ in 2001 to $0.34 \mu\text{g m}^{-2} \text{s}^{-1}$ in 2015-2017, and they correlate well with the observed regionally-mean dust storm days ($R=0.75$, $p < 0.001$) (Figure 1e). When using dust emission days (defined as the daily dust emission flux greater than a critical value such as 1, 10, or $50 \mu\text{g m}^{-2} \text{s}^{-1}$) as an indicator for the comparison with the observations, the correlation between simulations and observations are similar (R ranging from 0.71 to 0.80; Figure S8).” (Lines 153-160) and “The values of 1 and $10 \mu\text{g m}^{-2} \text{s}^{-1}$ are selected according to the field observations of Shao et al.⁵¹, which showed the low end of observed dust emission flux for particles smaller than $8.4 \mu\text{m}$ in diameter mostly lies within $1\text{-}10 \mu\text{g m}^{-2} \text{s}^{-1}$. In this study, the model calculates the dust emission flux for particles in a larger size range (i.e., smaller than $20 \mu\text{m}$ in diameter), and thus in addition to the values of 1 and $10 \mu\text{g m}^{-2} \text{s}^{-1}$, a larger value (i.e., $50 \mu\text{g m}^{-2} \text{s}^{-1}$) is also used.” (Figure S5 captions)

7. Lines131 - Drivers of weakening dust storm activity in last two decades: I remain rather confused by the choice of drivers of dust storms in the manuscript. The authors do not explain why wind speed, leaf area index, soil moisture, and snow cover were used as drivers of dust storm activity? In addition, the rest of the natural factors such as temperature and precipitation are completely ignored compared to snow cover, why?

Reply: We are sorry for the confusion. We have added more explanations in the revised manuscript. The reason why we choose wind speed, leaf area index, soil moisture, and snow cover is that they are the factors which directly affect the dust emission. For the rest of the natural factors such as temperature and precipitation, their impacts on the dust emission are considered indirectly through their impacts on the four factors selected. Therefore, although we do not consider other natural factors such as temperature and precipitation directly, their impacts on the dust emission are reflected

in the impacts from these four factors. For the trends of dust activity, snow cover plays a negligible role and thus is neglected. We have added the following sentence in the revised manuscript: “**Therefore, the impacts of atmospheric circulation, precipitation, and temperature are all reflected in the effects of the three factors examined.**” (Lines 260-261). Please see our reply to your comment #3 in more details.

8. Lines131 - Drivers of weakening dust storm activity in last two decades: Dust storms are natural processes due to multiple factors and the effect of a single factor on dust emission fluxes alone is not sufficient to describe the drivers of dust storm activity. Multi-factor integrated analysis models or methods should be used. For example, the southern part of the Qinghai-Tibet Plateau, where wind speeds are high, has relatively low dust emissions.

Reply: We agree with the reviewer that “***Dust storms are natural processes due to multiple factors and the effect of a single factor on dust emission fluxes alone is not sufficient to describe the drivers of dust storm activity.***” We would like to note that our model does incorporate the impacts of various factors on the dust emission based on our best knowledge of dust emission process. These factors include soil properties, surface wind speed, vegetation cover, and soil moisture, as described in Methods.

We do not use multi-factor integrated analysis models or methods, and the reasons lie in two aspects. First, the impacts of aforementioned factors have been represented by the model in a physical manner, and through sensitive experiments, we can quantify the impacts of these factors on the dust emission. Second, although the impacts of other factors (such as temperature and precipitation that can also affect the dust emission but indirectly) can be investigated using multi-factor integrated analysis models or methods, these investigations are beyond the scope of this study, which is to quantify the impacts of the multiple factors that can directly affect the dust emission.

For the example given by the reviewer, we would like to mention that soil moisture and

snow cover are both much larger in the Qinghai-Tibet Plateau than in desert regions. Thus the dust emission in the Qinghai-Tibet Plateau is relatively low although surface wind speeds there are high. Our simulated dust emission regions in the southern Qinghai-Tibet Plateau are consistent with the observations. We have added a statement on this: “**The model also reproduces the dust emission regions over the southern Tibetan Plateau.**” (Lines 126-127). In general, the determining factors for the spatial variations of dust emission consists of not only meteorological factors (surface wind, soil moisture, snow cover) but also land surface characteristics (soil properties and vegetation cover). Both the simulated spatial and seasonal variations of the dust emission are consistent with those of dust storm days, indicating the model’s capability of reproducing the dust emission. We have added a note on this: “**The spatial and seasonal variations of dust emission are shaped by the climatic factors (surface wind speed, soil moisture, and snow cover) and land surface characteristics (vegetation cover and soil properties), as represented by the dust emission model (details in Text S1-S2 and Figures S6-S7). Therefore, these results indicate that with the various influencing factors incorporated, the model can reproduce the spatial and seasonal variations of dust emission.**” (Lines 134-139).

Finally, we would like to mention that the main focus of this study is to understand the interannual variations and trends of dust activity in the last two decades. Therefore, the focus of Section “Drivers of weakening dust storm activity in last two decades” is to examine the drivers for the interannual variations and trends of dust activity but not on the seasonal variations of dust activity.

9. The analysis of vegetation trends in Figures 2c and 2d is not very accurate, and the colors of the figure legends do not seem to match.

Reply: We thank the reviewer for the comment. We are sorry for confusion. We think it is because that we used a different color scheme in Figure 2c compared to other panels, which made the reviewer confused. We have revised the color scheme in Figure 2c to

make it identical to other panels of Figure 2. We also clearly describe the analysis of vegetation cover trends: “Vegetation cover, represented here using LAI, also caused significant change in the dust emission (Figures 2c and 2d). Vegetation greening (a term used to describe the increase in vegetation) reduced the dust emission in most regions of northern China and southern Mongolia. Dust emission was enhanced in a small part of the regions due to vegetation degradation (or browning), such as in the eastern Tarim Basin, the southeastern Mongolia and adjacent China-Mongolia border regions, and a small region of the Qaidam Basin.” (Lines 181-187).

10. The authors only analyzed vegetation, wind speed and soil moisture, and the contribution of all three to dust emissions was 100%, which is not very reasonable. Furthermore, I believe that dust emissions are a complex process and that the results obtained from single factor analysis are not sufficient to describe the main drivers of dust emissions and their contribution, therefore, a multi-factor integrated analysis would be indispensable indeed.

Reply: We thank the reviewer for the comment. We agree with the reviewer that it is not very reasonable to assume the contributions from the three factors are 100%. In fact, they are relative contributions compared to the total contribution due to these three factors. We have changed “the contributions” to “the relative contributions” throughout the manuscript. We would like to mention that although the three factors should not explain all of the dust emission trends, they can explain most of the variations and trends of the dust emission during 2001-2017 because the model incorporated with these factors reproduces the temporal and spatial variations of dust activity during 2001-2017 in East Asia. Other factors either affect the dust emission indirectly or contribute little to the variations and trends of the dust emission in large areas (Please also see our replies to your comments #1, 3, and 7).

We agree with the reviewer that “*dust emissions are a complex process and that the results obtained from single factor analysis are not sufficient to describe the main*

drivers of dust emissions and their contribution". However, we would like to clarify that in this study we aim to uncover the main drivers for the interannual variations and trends of dust emissions (not dust emissions themselves). To quantify the impact from one individual factor, we conduct the sensitive experiments using a dust emission model. By only varying one factor during 2001-2017 while keeping other factors constant as in 2001, we quantify the contributions of that factor to the variations and trends of the dust emission during 2001-2017.

The model we use has taken into account the multiple factors that determine the dust emission. The method of conducting "sensitive experiments" by varying one factor at a time is commonly used in atmospheric and climate research, such as the International Panel on Climate Change (IPCC) report (Eyring et al., 2021). In this study, our purpose is to quantify the impacts of the main factors which can directly affect the dust emission. We believe that it is sufficient for our purpose.

References:

Eyring, V., N. P. Gillett, K. M. Achuta Rao, R. Barimalala, M. Barreiro Parrillo, N. Bellouin, C. Cassou, P. J. Durack, Y. Kosaka, S. McGregor, S. Min, O. Morgenstern, and Y. Sun, 2021: Human Influence on the Climate System. In *Climate Change 2021: The Physical Science Basis. Contribution of Working Group I to the Sixth Assessment Report of the Intergovernmental Panel on Climate Change* [Masson-Delmotte, V., P. Zhai, A. Pirani, S. L. Connors, C. Péan, S. Berger, N. Caud, Y. Chen, L. Goldfarb, M. I. Gomis, M. Huang, K. Leitzell, E. Lonnoy, J. B. R. Matthews, T. K. Maycock, T. Waterfield, O. Yelek, R. Yu, and B. Zhou (eds.)]. Cambridge University Press, Cambridge, United Kingdom and New York, NY, USA, pp. 423-552, doi:10.1017/9781009157896.005.

11. How were the variation curves for the factors in Fig. 3 obtained?

Reply: Fig. 3 is obtained by calculating the change in total dust emission over Eastern Sources for each month from different model experiments (Table 1), as done for the annual change shown in Fig. 1f. We have added an explanation in the caption of Fig. 3: **"The monthly changes are derived by calculating the dust emission change for each**

month from different model experiments (Table 1), as done in Fig. 1f. Note that summing the monthly changes for all the months yields the annual change shown in Figure 1f.” (Lines 752-755)

12. The authors obtained that wind speed is the main contributor to dust storm emissions and that April is the season with the largest number of strong dust storm days and dust storm days (Fig. S4), but how is it understood that in Fig. S7, dust emission fluxes are significantly higher in autumn than in spring and summer? In addition, vegetation cover is relatively better in autumn than in winter and spring, but dust emissions are higher than in other seasons? How can this logic be understood?

Reply: We are sorry for the confusion. In the original manuscript, Figure S4 (now replaced by Figure S5) shows the seasonal variations of strong dust emission days and dust storms days averaged during 2001-2017 based on the baseline experiment, while Figure S7 (now Figure S12) shows the linear trends of dust emission flux during 2001-2017. They are not contradictory. In Figure S7 (now Figure S12), the trends of dust emission flux are much larger in magnitude for spring than for other seasons (winter, summer, and autumn). If the reviewer gets the impression that the trends of dust emission fluxes are significantly higher in autumn than in spring and summer, we think this might be due to the color scheme. Now we have revised the color scheme and made the figure clearer to view. Please see the new Figure S12. We also added the seasonal mean dust emission fluxes during 2001-2017 for your reference (Figure S11). As you can see from Figure S11, dust emission fluxes are significantly higher in spring than in other seasons, which is consistent with Figure S5.

We also provide a full description of the seasonal variations in dust emission flux and associated factors in Section “**Text S2: Seasonal variations of dust emission and relevant driving factors**”. As mentioned by the reviewer, vegetation cover is relatively better in autumn than in winter and spring. Along with lower vegetation cover, higher

surface wind speed and lower soil moisture have led to the significantly higher dust emission in spring than in autumn. In winter, as there is a considerable amount of snow cover, dust emission is suppressed and thus dust emission in winter is lower than in autumn, although vegetation cover is smaller in winter than in autumn.

13. Lines 175-191, The authors mention that soil wetness is caused by precipitation and evaporation, but in summer evaporation is generally relatively strong in semi-arid areas, so why is the soil moister in summer than in other seasons? Vegetation is better in the autumn; wind speeds are higher and dust emissions are most variable in the autumn. This confuses me, the logic between vegetation and wind speed is not clearly stated and although wind speed plays a greater role than vegetation, it is difficult to convince the reader of a single factor alone.

Reply: We thank the reviewer for the comment. We have included a full explanation on the seasonal variations of dust emission flux and associated factors in Text S2. For the logic in quantifying the impacts of a single factor, we have added more explanations on our methodology.

Soil wetness is mainly determined by the balance of precipitation and evaporation. Although evaporation is stronger in summer than in other seasons, precipitation is also larger in summer and the excess of precipitation over evaporation leads to a relatively higher soil moisture in summer (Figures S6b and S7). We have clarified this in Text S2: “Soil moisture is lowest during middle spring to early summer (i.e., April to June), and highest in August and September due to the accumulation of precipitation (Figures S6b and S7). In the dust source regions, where runoff is small, soil moisture is mainly determined by the balance between precipitation and evaporation (Figure S7). Although evaporation is largest in summer, precipitation is also largest in this season and the excess of precipitation over evaporation leads to the increase of soil moisture from June to August” (Supplemental Lines 136-143).

The seasonal variation of dust emission depends on the seasonal variations of all the relevant variables including surface wind speed, soil moisture, vegetation, and snow cover (Figure S6). In spring, dust emission is largest than in other seasons, corresponding to the largest surface wind speed, relatively low soil moisture, and low LAI in the season. In contrast, dust emission is lowest in summer, when surface wind speed is smallest, soil moisture is relatively high, and LAI is largest during the year. In autumn and winter, dust emission occurs at times. Although surface wind is larger in winter than in autumn, dust emission is smaller in winter due to largest snow cover in this season. Therefore, dust emission is larger in autumn than in winter, and autumn is the second largest dust emission season in the year. We have added these explanations in Text S2.

Seasonal variations of simulated dust emission are mostly consistent with the observations of dust storm days. Although the model overestimates the dust emission in late autumn and winter, it captures the strongest dust activity in spring and much weaker dust activity in other seasons. This demonstrates the model's ability in reproducing the seasonal variations of dust activity and the rationality of our methodology. The model's overestimation of dust emission in later autumn and winter may be partly due to the underestimation of snow cover in these months used in the model (Reichle et al., 2017a). We have added a discussion on this: “**Note that the model overestimates the dust emission from October to February, which may be partly due to the underestimation of snow cover in MERRA-2 reanalysis used in the model⁵⁹**” (Supplemental Lines 161-164)

Our methodology is based on the experiments using a physically-based dust emission model which includes multiple determining factors for the dust emission. In the baseline experiment, we include all the considered factors to simulate the historical evolution of East Asian dust emission (Table 1). The impacts of these factors on the interannual variations and trends of dust emission are separated by comparing the sensitivity experiments (Table 1) with the baseline experiment. In the sensitivity experiments, the

seasonal variations of all the factors are included, and only the interannual variations of a given factor are excluded. In the revised manuscript, we have added more explanations on the methodology: “Note that for the experiments with the factors fixed as in 2001, the seasonal variations of these factors are still considered and set to the values as in 2001, but the interannual variations of these factors are excluded. By comparing these experiments, we identified the individual contributions of surface wind, vegetation cover, and soil moisture to the interannual variations of dust activity in East Asia.” (Lines 542-547)

14. The authors should have given a discussion and analysis of the seasonal variation in drivers and the relationship between drivers and dust emissions, rather than simply describing the phenomenon.

Reply: We thank the reviewer for the comment. Following your suggestion, we have added a detailed description on the seasonal variations of dust emission and the driving factors in Text S2:

“Seasonal variations of dust emission and relevant driving factors

The seasonal variations of dust emission are determined by the four driving factors (surface wind speed, soil moisture, vegetation cover, and snow cover) that are all considered in the dust emission model (DuEMv1.0).

Surface wind speed is highest in spring and lowest in summer (Figure S6a). Soil moisture is lowest during middle spring to early summer (i.e., April to June), and highest in August and September due to the accumulation of precipitation (Figures S6b and S7). In the dust source regions, where runoff is small, soil moisture is mainly determined by the balance between precipitation and evaporation (Figure S7). Although evaporation is largest in summer, precipitation is also largest in this season and the excess of precipitation over evaporation leads to the increase of soil moisture from June to August. Leaf area index (LAI) shows a strong seasonal variation with the largest

values in July-August and the smallest values during November to March (Figure S6c). Snow cover shows the strongest seasonal variation, with a peak in January and nearly zero from May to September (Figure S6d).

For the seasonal variations of dust emission, it is highest in spring, corresponding to the largest surface wind speed, relatively low soil moisture (especially in April when soil moisture is lowest during the year), and low LAI in this season. In contrast, dust emission is lowest in summer, due to the smallest surface wind speed, the largest LAI, and relatively high soil moisture in this season. In autumn and winter, dust emission occurs at times. Although surface wind is stronger in winter than in autumn, dust emission is lower in winter due to much more snow cover in winter than in autumn. Of all the seasons, dust emission is the second largest in autumn.

Seasonal variations of simulated dust emission are mostly consistent with the observations of dust storm days. Although the model overestimates the dust emission in autumn and winter, it captures the strongest dust activity in spring and much weaker dust activity in other seasons. This demonstrates the model's ability in reproducing the seasonal variations of dust activity and the rationality of our methodology. Note that the model overestimates the dust emissions from October to February, which may be partly due to the underestimation of snow cover in MERRA-2 reanalysis used in the model⁵⁹ (Supplemental Lines 131-164).

Same as the annual mean, we also calculate the relative contributions from the changes in surface wind speed, vegetation cover, and soil moisture to the trends of dust activity for each season and summarize the results in Table S2:

Table S2. Dust emission over Eastern Sources for annual total (ANN; unit: Tg yr⁻¹) and each season (MAM, JJA, SON, and DJF; unit: Tg season⁻¹) during 2001 and 2010-2017, as well as the changes from 2001 to 2010s (2010-2017) simulated by the four experiments listed in Table 1.

Exp. name	2001					2010-2017				
	ANN	MAM	JJA	SON	DJF	ANN	MAM	JJA	SON	DJF
Baseline (All: Wind+LAI+SOILM)	310	198	34	24	54	202	103	26	39	34
Wind	310	198	34	24	54	254	131	37	46	40
LAI	310	198	34	24	54	273	177	27	20	49
SOILM	310	198	34	24	54	281	178	28	25	50
	Changes from 2001 to 2010s (2010-2017)					Relative contributions to the total changes (%) ^a				
	ANN	MAM	JJA	SON	DJF	ANN	MAM	JJA	SON	DJF
Baseline (All: Wind+LAI+SOILM)	-108	-95	-8	15	-20	100%	100%	100%	100%	100%
Wind	-56	-67	3	22	-14	46%	62%	-30% ^b	116%	61%
LAI	-37	-21	-7	-4	-5	30%	19%	70%	-21% ^b	22%
SOILM	-29	-20	-6	1	-4	24%	19%	60%	5%	17%

^a: The relative contribution from each factor (Wind, LAI, SOILM) is calculated by dividing the individual change due to the factor to the sum of all the individual changes due to the three factors. For example, 46% is derived by dividing -56 by -122 (=-56+(-37)+(-29)).

^b: The negative contributions indicate that the change caused by an individual factor is opposite in sign to the total change by all the three factors.

Finally, in the revised manuscript, we added a discussion on the seasonal variations in drivers and the relationship between drivers and dust emissions: “The relative contributions of each factor to the total changes are summarized in Table S2. In spring, the changes in surface wind, vegetation cover, and soil moisture account for 62%, 19%, and 19%, respectively, of the total change. In winter, the relative contributions from these three factors are similar to those in spring. In contrast, the changes in both vegetation cover and soil moisture contribute more to the total changes than surface wind speed in summer, when surface wind speed is lowest during the year.” (Lines 230-

235).

15. Lines 203-204: Only 23% of dust emission reduction from 2003 2001 to the 2010s in grasslands were a result of vegetation greening. Where do I get it from?

Reply: Please see Figure 4d, which illustrates the reduction of dust emission (Tg yr^{-1}) during 2010-2017 relative to 2001 over grasslands due to wind, LAI, soil moisture (SOILM), and all these three factors, respectively. We have noted Figure 4d when mentioning the value of 23% (Line 250).

Reviewer #3 (Remarks to the Author):

I thank the authors for their extensive revision and responses to my previous comments. My original comments regarding the comparison with Tai et al. (2021) were intended to ensure the authors would acknowledge all past, especially the recent, work that examined the same issue, giving credits to them and carefully explaining the new aspects, possible improvements and new insights offered by their own work in the context of previous work. They were not asked to be the judge to give a definitive statement on which is "better", which would be an unfair conclusion either way due exactly to the differences in assumptions and methodologies employed. However, the authors have tried extensively to discredit the previous work and for many times stated that their work was far "better", exaggerating the novelty of their own work that is simply not grounded in their methods, results and conclusions. Such an overcritical approach also undermines the scientific objectivity and professionalism of their manuscript. Therefore, I do not recommend the publication of this manuscript unless the authors can revise the manuscript, taking a more objective and comparative approach, giving full credit to and acknowledging the significance of previous work, and simply highlighting what they have done differently and how their work offers new insights that the previous work may not convey.

Reply: We thank the reviewer for expressing his/her concerns. In the previously revised manuscript, we tried to compare our study with Tai et al. (2021) as we would like to make the readers better aware of the differences between our study and Tai et al. (2021). We agree with the reviewer that we should not provide a definitive judgement on which work is “better”, due to the differences in assumptions and methodologies employed. Following the reviewer’s comment, we have removed the statements on which is “better”, taken a more objective and comparative approach, and given full credit to and acknowledged the significance of previous work in the revised manuscript.

To give full credit to and acknowledge the significance of Tai et al. (2021) and others,

we have highlighted the previous studies: “Several studies have used models to understand the long-term variations of East Asian dust^{8,19-21}, and two of them have further separated the contributions of climatic factors and vegetation cover^{19,21}.” (Lines 64-66). We also highlight the key findings of Tai et al. (2021): “Tai et al. showed that the interannual variability and trends of East Asian dust emission are largely shaped by climatic factors, with the vegetation cover playing a secondary but locally important role especially in the semiarid and non-desert regions undergoing rapid land cover change and desertification; Of all the climatic factors, surface wind speed, followed by total precipitation, was the most important meteorological factor controlling dust variability and trends, with the weakening of wind playing the largest role in the overall decline of dust emissions in Taklimakan Deserts and Gobi Deserts during 1982-2010²¹.” (Lines 68-75).

Following the reviewer’s suggestion, we simply highlight what we have done differently and how our work offers new insights that the previous work may not convey: “These two studies focused on the period of 1982 to 2006/2010, and it is still unclear about the roles of climatic factors and vegetation cover in the last two decades when more intensive ecological restorations were implemented¹⁵.” (Lines 75-78), “We focus on dust emission which is the first and foremost component of the dust cycle.” (Lines 82-83), and “A recent study has adopted a similar approach to isolate the impacts of climatic factors and vegetation cover²¹. In that study, the impacts of soil moisture and surface wind were considered together. Our study makes a step forward by separating the impacts of these two factors.” (Lines 549-552).

We also add a discussion on the key differences of the conclusions between Tai et al. (2021) and our study: “Note that previous study of Tai et al.²¹ showed that vegetation cover induces much smaller impacts on the long-term variability and trends of dust activity (particularly in Gobi Deserts) compared to climatic factors during 1982-2010. Our study shows larger impacts from vegetation cover change compared to that study, which can partly be ascribed to more intensive change of vegetation cover during

2000s-2010s in our study compared to the earlier decades in that study¹⁵. Larger impacts in our study may also result from the differences in the dust emission parameterizations and meteorological forcings (Table S4). For example, our model considers the reduction of erodible surface fraction due to increasing LAI, which reduces the dust emission, while bare soil fraction is set to be constant in that previous study (Table S4).” (Lines 276-285).

After these revisions, we believe we have fully acknowledged the previous study of Tai et al. (2021), presented an objective comparison between that study and our study, and provided a clear interpretation of new insights offered by our study.

More specifically:

"A recent study has attempted to quantify the impacts of climatic factors and vegetation cover on dust emission variations over East Asia during 1982-2010, but the results remain questionable as the model they used is not fully validated. Compared to the observations of aerosol optical depth in 2000- 2010 excluding 2001, the model shows low fidelity in the temporal variations of dust activity."

- The model the previous work used was adequately validated using multiple datasets that are more varies and comprehensive than the authors' work. For instance, this current study only used dust emission observations, which are extremely sparse, as well as dust storms, which are more comprehensive, whereas the previous work used a variety of data for dust storms, concentrations and optical depths. This is not a limitation of this work, though, because this work focuses only on dust emissions while the previous work focused on the whole atmospheric cycle of dust. Due to the different scopes of the studies, it is unnecessary and rather irrelevant to erroneously discredit the previous work to "call for a more reliable quantitative assessment with an improved model". A due discussion of the previous work, highlighting the key findings and significance, while stating the different aspects and perspectives that their own work can bring, is enough.

Reply: We thank the reviewer for pointing out this. In the previously revised manuscript, we mentioned the possible limitations of the previous study of Tai et al. (2021) as we would like to make the readers aware of the differences between our study and Tai et al. (2021). To address the reviewer's concern, we have removed the sentences in the previous manuscript: *“A recent study has attempted to quantify the impacts of climatic factors and vegetation cover on dust emission variations over East Asia during 1982-2010, but the results remain questionable as the model they used is not fully validated. Compared to the observations of aerosol optical depth in 2000- 2010 excluding 2001, the model shows low fidelity in the temporal variations of dust activity.”* and *“Limitations in that study call for a more reliable quantitative assessment with an improved model that can reproduce dust variations shown in the observations”*.

Following the reviewer's comment, we have highlighted the key findings and significance of Tai et al. (2021): *“Tai et al. showed that the interannual variability and trends of East Asian dust emission are largely shaped by climatic factors, with the vegetation cover playing a secondary but locally important role especially in the semiarid and non-desert regions undergoing rapid land cover change and desertification; Of all the climatic factors, surface wind speed, followed by total precipitation, was the most important meteorological factor controlling dust variability and trends, with the weakening of wind playing the largest role in the overall decline of dust emissions in Taklimakan Deserts and Gobi Deserts during 1982-2010²¹.”* (Lines 68-75).

We also agree with the reviewer to state the different aspects of our study. Compared to the study of Tai et al. (2021), our study focuses on the more recent period with more intensive change of vegetation cover (Chen et al., 2019): *“These two studies focused on the period of 1982 to 2006/2010, and it is still unclear about the roles of climatic factors and vegetation cover in the last two decades when more intensive ecological restorations were implemented¹⁵.”* (Lines 75-78). Our study also focuses on dust emission while Tai et al. (2021) focused on the whole atmosphere cycle of dust. We also add an explanation on the importance of our focus: *“We focus on dust emission*

which is the first and foremost component of the dust cycle.” (Lines 82-83).

In addition, we also mention the difference in the methodology: “A recent study has adopted a similar approach to isolate the impacts of climatic factors and vegetation cover²¹. In that study, the impacts of soil moisture and surface wind were considered together. Our study makes a step forward by separating the impacts of these two factors.” (Lines 549-552).

"The model is validated with dust storm records (Figure S2) in terms of spatial and temporal variations and thus provides a reliable estimation of dust emissions."

- There is no guarantee that dust storm activities provide more "reliable" evidential basis than other dust parameters such as dust optical depths or concentrations. This statement on its own is simply not necessary.

Reply: We thank the reviewer for pointing out this. Following the reviewer’s comment, we have removed the statement “*and thus provides a reliable estimation of dust emissions*”.

"Our study has great advantages over it in terms of the modeling framework, including dust emission parameterizations to account for the impacts of vegetation cover, meteorology forcing, and input data of LAI (Text S2)"

- Again, the authors were not asked to be the judge, and indeed those "great advantages" claimed are rather dubious. I fully acknowledge that the emission model of dust per se has several aspects that more comprehensively represent the physical dependence of dust emissions on various factors. But comprehensiveness of complexity of model processes do not automatically lend the model greater credence, especially without long-term observations of the direct physical outcomes to validate (i.e., dust emission flux). Also, the authors fail to acknowledge that a key aspect of Tai et al. (2021) that their study did not address is the atmospheric transport and deposition of dust as simulated in a full chemical transport model, which for sure

introduce more uncertainties due to the larger scope involved but can then make use of a larger variety of datasets to validate model outcomes. Such key outcomes and the major conclusions therein were not discussed at all by the authors, who have opted to focus on unfairly discrediting the previous work.

Reply: We thank the reviewer for the comment. We did not intend to discredit the work of Tai et al. (2021), and we are sorry that we did not acknowledge the advantage of Tai et al. (2021) in simulating the whole atmospheric cycle of dust (e.g., emission, transport and deposition). Following the reviewer's comment, we have removed the statement in the revised manuscript: *“Our study has great advantages over it in terms of the modeling framework, including dust emission parameterizations to account for the impacts of vegetation cover, meteorology forcing, and input data of LAP”*.

We have discussed the key outcomes and the major conclusions of Tai et al. (2021) in Introduction of the revised manuscript: *“Tai et al. showed that the interannual variability and trends of East Asian dust emission are largely shaped by climatic factors, with the vegetation cover playing a secondary but locally important role especially in the semiarid and non-desert regions undergoing rapid land cover change and desertification; Of all the climatic factors, surface wind speed, followed by total precipitation, was the most important meteorological factor controlling dust variability and trends, with the weakening of wind playing the largest role in the overall decline of dust emissions in Taklimakan Deserts and Gobi Deserts during 1982-2010²¹”* (Lines 68-75).

We also have explained the focus of our study in Introduction: *“We focus on dust emission which is the first and foremost component of the dust cycle”* (Lines 82-83) and acknowledged the importance of dust transport and deposition by citing the Tai et al. study: *“On the other hand, due to the impacts of dust transport and deposition, the benefits of reduced dust emission in alleviating dust pollution may vary in different regions and should be assessed with the full consideration of dust transport and*

deposition²⁰⁻²³.” (Lines 313-316).

"The results in this study are more reliable than those of Tai2021"

- This is simply not grounded by evidence, and is indeed an unnecessary comparison to make to begin with. As stated by the authors themselves, the long-term observations used that are the most comprehensive are dust storm days, but the physical connections between dust emission per se and dust storm activity are far from immediate - dust storm activity can be further modulated by other processes such as transport and deposition, which were not considered by the authors' model but considered by Tai et al. (2021). Understandably, different models have different functions, aims and uncertainties, and saying one is more reliable than other, especially in this case, is like comparing oranges to apples.

Reply: We thank the reviewer for the comment. Following the reviewer's comment, we have removed the section Text S2 (Comparison of this study with a recent study of Tai et al. (2021)) including the statement: *"The results in this study are more reliable than those of Tai2021"*.

"On the other hand, due to the impacts of dust transport and deposition, the benefits of reduced dust emission in alleviating dust pollution may vary in different regions and should be assessed with full consideration of dust transport."

- This is exactly an "advantage" of Tai et al. (2021), but the authors fail to acknowledge this.

Reply: We thank the reviewer for the comment. We have cited Tai et al. (2021) for the statement in the revised manuscript: ***"On the other hand, due to the impacts of dust transport and deposition, the benefits of reduced dust emission in alleviating dust pollution may vary in different regions and should be assessed with the full consideration of dust transport and deposition²⁰⁻²³."*** (Lines 313-316).

REVIEWER COMMENTS

Reviewer #1 (Remarks to the Author):

Thanks very much to the author for responding carefully to my previous comments on the manuscript and for giving careful revisions. I believe the manuscript can be considered for publication after discussing the following minor changes.

- 1. Dust storms are a natural process caused by a combination or interaction of several factors, so I think that if possible I would suggest that the authors use a relevant approach to investigate the combination or interaction of drivers and their contribution to the dust emission process. For example, the main conclusion of the manuscript is that the reduction in dust emissions is a result of the weakening of surface winds, the increase in vegetation cover and soil moisture, and that ecological restoration projects have to some extent also influenced the weakening of surface winds, which in turn has indirectly reduced dust emissions. However, there are interactions between these factors and together they contribute to dust emissions, therefore, I think it is more important to explore the trends of multiple factors on dust emissions.**
- 2. Line 712 Figure resolution needs to be improved, and attention should be paid to consistency in figure names, fonts, etc.**
- 3. Lines 167-168 The effect of snow cover on dust emission trends is negligible due to its very small variability, why does it need to be considered in modelling?**
- 4. For the section on Drivers of the weakening dust activity in last two decades, the single factor driven analysis does not fully indicate the mechanism of the reduction in dust emissions and I would suggest that the authors add methods such as classification and regression trees to the driven analysis if possible.**
- 5. Lines 255-264 should be part of the discussion section.**
- 6. In the discussion section, the discussion of seasonal trends in dust emissions is not sufficient and the authors should add to this section with previous studies. For example, most previous studies have shown that dust emissions in northwest China are significantly higher in spring and winter than in summer and autumn, but the higher dust emissions in autumn in this manuscript are not entirely consistent with previous studies. Therefore, it is recommended that the authors provide an analysis of the mechanisms underlying the seasonal trends in dust emissions in this manuscript in the context of previous studies (possible to add in the supplementary information).**

Reviewer #3 (Remarks to the Author):

The authors have adequately addressed my previous concerns. I recommend publication of the paper.

REVIEWER COMMENTS

Reviewer #1 (Remarks to the Author):

Thanks very much to the author for responding carefully to my previous comments on the manuscript and for giving careful revisions. I believe the manuscript can be considered for publication after discussing the following minor changes.

Reply: We thank the reviewer for his/her constructive reviews and comments. We have revised the text and figures following your suggestions.

1. Dust storms are a natural process caused by a combination or interaction of several factors, so I think that if possible I would suggest that the authors use a relevant approach to investigate the combination or interaction of drivers and their contribution to the dust emission process. For example, the main conclusion of the manuscript is that the reduction in dust emissions is a result of the weakening of surface winds, the increase in vegetation cover and soil moisture, and that ecological restoration projects have to some extent also influenced the weakening of surface winds, which in turn has indirectly reduced dust emissions. However, there are interactions between these factors and together they contribute to dust emissions, therefore, I think it is more important to explore the trends of multiple factors on dust emissions.

Reply: We thank the reviewer for pointing out this. We agree with the reviewer that dust storms are a natural process caused by a combination or interactions of multiple factors. To quantify the contributions of different factors to the declining trends of dust activity in East Asia during 2001-2017, we use a physically-based dust emission model and compare the simulation results with each of influencing factors set to the values as in the beginning year (2001). This approach is a step forward to isolate the impacts of multiple factors that directly affect the dust emission, including surface wind speed, vegetation cover, and soil moisture. The results are easy to interpret.

Meanwhile we also acknowledge the limitation of our approach. As pointed out by the reviewer, there are interactions between the factors which can affect dust emission, and the impacts of these interactions on dust emission are not investigated in the study. In fact, MERRA-2 does incorporate the potential interaction between surface wind and soil moisture, but it adopts climatological vegetation parameters (leaf area index and greenness), which only shows seasonal variations but does not vary for different years (Reichle et al., 2017). Therefore, MERRA-2 does not directly incorporate the impacts of vegetation cover change (including the change induced by ecological restorations) on both surface winds and soil moisture, although the observations assimilated into MERRA-2 may include the influence of vegetation cover to some extent. However, it is difficult to distinguish these effects at the present time, as MERRA-2 data is already frozen.

In the revised manuscript, we have discussed the interactions between multiple factors, and acknowledged the limitation of our approach: “Surface winds, vegetation cover, and soil moisture are interrelated to each other. For example, the increase in vegetation cover may decrease the surface winds, which also contributes to the decline of dust activity. Some of this effect may be included in the MERRA-2 data, as the observations are assimilated into the MERRA-2 reanalysis. However, it is difficult to quantify the effect of this interaction on dust emission as the MERRA-2 data are now frozen and no further adjustment is available.” (Lines 289-294)

To address this issue, it is desirable to use climate models or Earth System Models (ESMs) which can simulate the atmosphere-land-vegetation interactions and their impacts on dust emission. However, such a model has a high degree of freedom and is subject to large uncertainty. For example, none of up-to-date climate models and ESMs are able to capture the strong decline of dust activity in last decades (Wu et al., 2018). Therefore, at the present time, a dust emission model is still a viable approach for the attribution of dust activity change to various factors.

In the revised manuscript, we have added a discussion on the potential benefits and disadvantages of using climate models and ESMs: “Ideally, this limitation can be overcome with a more complex model, such as climate models and Earth System Models (ESMs) which can represent the atmosphere-land-vegetation interactions^{8,57}. However, none of the climate models and ESMs to date can reproduce the strong decline of East Asian dust activity during the past decades⁸. Further developments in the forcing data and climate models/ESMs are needed to quantify the impacts of atmosphere-land-vegetation interactions on dust emission.” (Lines 294-300).

References:

Reichle, R. H., C. S. Draper, Q. Liu, M. Girotto, S. P. P. Mahanama, R. D. Koster, and G. J. M. De Lannoy (2017), Assessment of MERRA-2 Land Surface Hydrology Estimates, *Journal of Climate*, 30(8), 2937-2960, doi:10.1175/jcli-d-16-0720.1.

2. Line 712 Figure resolution needs to be improved, and attention should be paid to consistency in figure names, fonts, etc.

Reply: Thank you for the comment. We have replotted Figure 1 to improve its resolution. We also revised the figure names, fonts, and texts to make them consistent with each other. Please see the new Figure 1.

3. Lines 167-168 The effect of snow cover on dust emission trends is negligible due to its very small variability, why does it need to be considered in modelling?

Reply: Thank you for the comment. Snow cover can directly affect the fraction of exposed bare soil and thus needs to be considered in modeling (Text S1). When we mentioned the effect of snow cover on dust emission trends are negligible due to its small variability, we are referring to the interannual variations and temporal trends during 2001-2017. For the seasonal variations of dust emission, the impacts of snow

cover are significant and snow cover is one of the key factors that determine the seasonal variations of dust activity (Text S3 and Figures S6-S7).

In the revised manuscript, we have clarified: “**Apart from snow cover, which shows small impacts on the trends of dust emission due to its small change over the main dust source regions during 2001-2017 (Figure S11), other factors contribute significantly to the decline of dust emission in the period from 2001 to 2017 (Figure 2).**” (Lines 170-173). We also mentioned the potential underestimation of snow cover impacts on dust emission due to the discrepancy in MERRA-2 data: “**Note that as snow cover fraction is underestimated in MERRA-2 compared to observations, the impacts of snow cover on dust emission may be underestimated in the simulations especially in the cold season⁵⁹.**” (Supplemental Lines 121-123)

4. For the section on Drivers of the weakening dust activity in last two decades, the single factor driven analysis does not fully indicate the mechanism of the reduction in dust emissions and I would suggest that the authors add methods such as classification and regression trees to the driven analysis if possible.

Reply: Thank you for the comment. We would like to mention that although the single factor driven analysis does not fully indicate the mechanism of the reduction in dust emissions, it is an effective approach to quantify the contributions of various factors which can directly affect the dust emissions. Our approach is based on a physically-based dust emission model and the conclusions are obtained by comparing the simulation results from multiple sensitivity experiments with the baseline model results. Such an approach has been widely used in climate and environment research (e.g., Gillett et al., 2016; Zhang et al., 2019) and can provide valuable insights into the attribution of dust activity change, as shown in this study. Our analysis may not fully indicate the mechanism of dust emission reductions because of the limitations in the forcing data and the dust emission model used. Please also see our reply to your comment #1 on the role of the interactions between multiple factors in the dust emission

reduction.

For the classification and regression trees (CART), it is a predictive algorithm and has generally been used to predict the target variable based on certain predictor variables (Loh, 2011; Krzywinski and Altman, 2017). CART or CART-based methods have been adopted in dust-related research, such as to determine the key factors for the occurrence and intensity of dust activity (Guan et al., 2017; Ebrahimi-Khusfi et al., 2021). In our study, since our model has already represented the dust emission based on the dust emission equation (Text S1), we will not use other methods such as CART to predict the occurrence and intensity of dust emission. However, we will use the CART method in our future analysis of dust activity.

References:

- Ebrahimi-Khusfi, Z., Nafarzadegan, A. R., Dargahian, F., Predicting the number of dusty days around the desert wetlands in southeastern Iran using feature selection and machine learning techniques, *Ecological Indicators*, 125, 107499, <https://doi.org/10.1016/j.ecolind.2021.107499>, 2021.
- Gillett, N. P., Shiogama, H., Funke, B., Hegerl, G., Knutti, R., Matthes, K., Santer, B. D., Stone, D., and Tebaldi, C.: The Detection and Attribution Model Intercomparison Project (DAMIP v1.0) contribution to CMIP6, *Geosci. Model Dev.*, 9, 3685–3697, <https://doi.org/10.5194/gmd-9-3685-2016>, 2016.
- Guan, Q., Sun, X., Yang, J., Pan, B., Zhao, S., & Wang, L.. Dust Storms in Northern China: Long-Term Spatiotemporal Characteristics and Climate Controls, *Journal of Climate*, 30(17), 6683-6700, 2017.
- Loh, W.-Y., Classification and regression trees. *WIREs Data Mining Knowl Discov*, 1: 14-23. <https://doi.org/10.1002/widm.8>, 2011.
- Krzywinski, M., Altman, N. Classification and regression trees. *Nat Methods* 14, 757–758. <https://doi.org/10.1038/nmeth.4370>, 2017.
- Zhang, Q., et al., Drivers of improved PM_{2.5} air quality in China from 2013 to 2017, 116(49), 24463-24469, doi:10.1073/pnas.1907956116, *J*

5. Lines 255-264 should be part of the discussion section.

Reply: Thank you for the suggestion. We have moved these sentences to the Discussion section (Lines 261-271).

6. In the discussion section, the discussion of seasonal trends in dust emissions is not sufficient and the authors should add to this section with previous studies. For example, most previous studies have shown that dust emissions in northwest China are significantly higher in spring and winter than in summer and autumn, but the higher dust emissions in autumn in this manuscript are not entirely consistent with previous studies. Therefore, it is recommended that the authors provide an analysis of the mechanisms underlying the seasonal trends in dust emissions in this manuscript in the context of previous studies (possible to add in the supplementary information).

Reply: We thank the reviewer for the careful review and pointing out this to us. We have added more analysis and discussions on the seasonal variations of dust storm days and dust emissions.

As pointed out by the reviewer, most previous studies have shown that dust activity (in terms of dusty days, dust storm days, dust storm frequency, or dust event frequency) in northern China is significantly higher in spring and winter than in summer and autumn (e.g., Guan et al., 2015, 2017; Zhao et al., 2018).

In our study, the number of dust storm days from observations is also higher in winter than in autumn (Figures S3 and S6). The difference between our study and previous studies is that the contrast between winter and autumn in our study is not as large as in previous studies (Figures S3 a-b). The reason for the smaller contrast in our study is

that we focus on the past two decades while these previous studies (Guan et al., 2015, 2017; Zhao et al., 2018) extended their study periods to earlier decades such as 1960-2007 or 1982-2013. More specifically, compared to previous decades (1960s-1970s or 1980s-1990s), the smaller contrast of dust activity between winter and autumn in 2001-2017 results from the fact that dust activity has decreased more strongly in winter than in autumn, as shown in Figure S3 c-d.

For the simulations, our model does show significantly stronger dust emission in spring than in other seasons and weakest dust emissions in late summer and early autumn, which is consistent with the observations. For the whole autumn season, the reviewer pointed out that our model simulates higher dust emission in this season than in winter. However, we note that this feature is only present for the entire Eastern Sources (Figure S6e). At the synoptic stations, the model simulates larger dust emission flux and more days with dust emission flux >1 , 10, or $50 \mu\text{g m}^{-2} \text{ s}^{-1}$ in winter than in autumn (Figure S6 a-d). This is generally consistent with the observations.

Since these synoptic stations are located in the Chinese part of Eastern Sources, we further separate the dust emission of Eastern Sources into Chinese part (ES-CN) and Mongolian part (ES-MG), respectively. As shown in Figure R1, the model simulates similar seasonal variations of dust emission over ES-CN and ES-MG, but over ES-MG it simulates higher dust emission in autumn than in winter, while over ES-CN the model simulates lower dust emission in autumn than in winter. The difference of dust emission between winter and autumn can also be seen from Figure S13, which shows that the simulated dust emission in southern Mongolia is significant higher in autumn than in winter. Although the synoptic observations in Mongolia are unavailable for use in the model evaluation, according to the study of Natsagdorj et al. (2003) using synoptic observations during 1960-1989, a second maximum of dust storm frequency over Mongolia is in autumn following the highest frequency in spring, and the minimum frequency is in summer. Our model results in Mongolia are generally consistent with the observations in Natsagdorj et al. (2003).

In the revised manuscript, we have clarified the seasonal variations of dust activity from observations: “Dust activity also shows large seasonal variations, with dust storms occurring most frequently in spring and least frequently in late summer and early autumn. This result is generally consistent with previous studies^{5-6,11}, but our study shows a smaller contrast in dust activity between winter and autumn due to the fact that we focus on more recent decades (Text S2).” (Lines 102-106).

We also added more analysis and discussions on the seasonal variations of dust storm days in supplement Text S2:

Text S2

Seasonal variations of dust storm days and comparison with previous studies

During 2001-2017, dust storm occurs most frequently in spring and least frequently in late summer and early autumn. Dust storm also occurs at times in winter and late autumn, with slightly more dust storm days in late winter than in early winter and late autumn (Figure S3).

Most previous studies showed that dust activity in Gobi Deserts and Sandy Lands of northern China is much stronger in spring than in other seasons^{5-6,11}. Our results are consistent with them. Previous studies also showed a second peak of dust activity in winter, which is significantly stronger than that in autumn^{5-6,11}. Compared to these studies, the contrast of dust activity between winter and autumn in our study is smaller, which can be ascribed to the difference in the study periods between our study and these previous studies. Our study focuses on the past two decades, while most previous studies extended the study periods to earlier decades (Figure S3a). Based on a continuous dataset during 1954-2017 (see Methods), dust activity is significantly weaker for all the four seasons during 2001-2017 than in the previous decades (1960s-1970s and 1980s-1990s), with the most significant decline in spring and winter (Figure S3 c-d). Therefore, the stronger decline of dust activity in winter than in autumn in past several decades results in a smaller contrast in the dust storm days between these two

seasons during 2001-2017.

For the model's overestimation of dust emission in autumn and winter, we also added more explanations: "The overestimation of dust emission in autumn and winter can also be ascribed to the fact that the model does not consider the impacts of dead leaves on dust emission. A previous study showed that dead leaves can serve as a major type of roughness elements after the growing season and effectively reduce the dust emission in East Asia⁷¹." (Supplemental Lines 189-193).

As for the drivers of the seasonal trends in dust emissions, we have provided an analysis of simulation results in Table S2 and Text S2 (now Text S3) of the previous version of the manuscript. Now, following the reviewer's suggestion, we have added more discussions on these results in the Discussion section: "East Asian dust activity has large seasonal variations and is much stronger in spring than in other seasons. The dust emission reduction in spring accounts for most of the annual total reduction, followed by the reduction in winter (Figure 3; Table S2). Surface winds dominate the dust emission reduction in spring and winter, but changes in surface winds tend to increase the dust emission in autumn and summer. In contrast, changes in vegetation cover and soil moisture lead to the reduction of dust emission in almost all the months (Figure 3). Although change in either vegetation cover or soil moisture accounts for around 20% of total dust emission reduction by the three factors (surface winds, vegetation cover, and soil moisture) in spring and winter, they play a key role in the dust emission reduction in summer (Table S2). In autumn, the dust emission reduction induced by vegetation cover greenness cancels out part of the dust emission increase by surface winds. While most of previous studies focused on the dust variations for the spring season^{7,19-21,30} or the annual mean^{5-6,8,11,18}, our study shows large variations in the trends of dust emission and associated surface winds during the four seasons. The mechanisms for these variations deserve further investigation." (Lines 273-287).

Figure R1. Seasonal variations of regionally accumulated dust emission over Chinese and Mongolian part of Eastern Sources, respectively.

References:

- Guan, Q., J. Yang, S. Zhao, B. Pan, C. Liu, D. Zhang, and T. Wu (2015), Climatological analysis of dust storms in the area surrounding the Tengger Desert during 1960–2007, *Climate Dynamics*, 45(3), 903-913, doi:10.1007/s00382-014-2321-3.
- Guan, Q., X. Sun, J. Yang, B. Pan, S. Zhao, and L. Wang (2017), Dust Storms in Northern China: Long-Term Spatiotemporal Characteristics and Climate Controls, *J Climate*, 30(17), 6683-6700, doi:10.1175/jcli-d-16-0795.1.
- Natsagdorj, L., D. Jugder, and Y. S. Chung (2003), Analysis of dust storms observed in Mongolia during 1937-1999, *Atmos Environ*, 37(9-10), 1401-1411, doi:Doi 10.1016/S1352-2310(02)01023-3.
- Zhao, Y., Z. Xin, and G. Ding (2018), Spatiotemporal variation in the occurrence of sand-dust events and its influencing factors in the Beijing-Tianjin Sand Source Region, China, 1982–2013, *Regional Environmental Change*, 18(8), 2433-2444, doi:10.1007/s10113-018-1365-z.

Reviewers' Comments:

Reviewer #1:

Remarks to the Author:

According to the previous review comments, the author has carefully revised this manuscript "Drivers of recent decline in dust activity over East Asia: The roles of climatic factors and ecological restoration". I think the manuscript could be published in the journal of "Nature Communications".

Reviewer #1 (Remarks to the Author):

According to the previous review comments, the author has carefully revised this manuscript “Drivers of recent decline in dust activity over East Asia: The roles of climatic factors and ecological restoration”. I think the manuscript could be published in the journal of “Nature Communications”.

Reply: We thank the reviewer for the positive comments and for recommending the publication of our paper in *Nature Communications*.